

# Algebraic theory of quantum synchronization and limit cycles under dissipation

**Berislav Buča[1*], Cameron Booker[1] and Dieter Jaksch[1, 2]**

**1** Clarendon Laboratory, University of Oxford, Parks Road, Oxford OX1 3PU, United Kingdom
**2** Institut für Laserphysik, Universität Hamburg, 22761 Hamburg, Germany

* berislav.buca@physics.ox.ac.uk

## Abstract

Synchronization is a phenomenon where interacting particles lock their motion and display non-trivial dynamics. Despite intense efforts studying synchronization in systems without clear classical limits, no comprehensive theory has been found. We develop such a general theory based on novel necessary and sufficient algebraic criteria for persistently oscillating eigenmodes (limit cycles) of time-independent quantum master equations. We show these eigenmodes must be quantum coherent and give an exact analytical solution for *all* such dynamics in terms of a dynamical symmetry algebra. Using our theory, we study both stable synchronization and metastable/transient synchronization. We use our theory to fully characterise spontaneous synchronization of autonomous systems. Moreover, we give compact algebraic criteria that may be used to prove *absence* of synchronization. We demonstrate synchronization in several systems relevant for various fermionic cold atom experiments.



# 1 Introduction to Quantum Synchronization

Understanding complex dynamics is arguably one of the main goals of all science from biology (e.g. [1, 2]) to economics (e.g. [3, 4]). Synchronization is a remarkable phenomenon where multiple bodies adjust their motion and rhythms to match by mutual interaction. It is one of the most ubiquitous behaviours in nature and can be found in diverse systems ranging from simple coupled pendula, neural oscillations in the human brain [5], epidemic disease spreading in the general population, power grids and many others [6]. Understanding this phenomenon has been one of the major successes of dynamical systems theory and deterministic chaos [7, 8].

Intense work in the last decade has extended these non-linear results to the semi-classical domain of certain quantum systems with an infinite or very large local Hilbert space, e.g. quantum van der Pol oscillators, bosons, or large spin-$S$ systems [9–31]. These systems are usually understood successfully through mean-field methods or related procedures that neglect the full quantum correlations. By contrast, strongly interacting systems that have finite-dimensional local Hilbert spaces, such as finite spins, qubits or electrons on a lattice, can most often not be adequately treated with mean-field methods [32] and semi-classical limits cannot be directly formulated for them [33–35]. For the sake of brevity, we will call such systems "quantum" and the previous ones "semi-classical". Very recently, these systems with finite-dimensional local Hilbert spaces have attracted much attention as possible platforms for quantum synchronization [36–49]. Crucially, the subsystems that are to be synchronized in such quantum systems do not have easily accessible $\hbar \rightarrow 0$ or equivalent large parameter (semi-classical) limits.

Quantum synchronization holds promise for technical applications. For instance, synchronizing spins in a quantum magnet would allow for homogeneous and coherent time-dependent magnetic field sources. Developing sources of homogeneous and coherent time-dependent magnetic fields has the significant potential to improve the resolution of MRI images [50]. Additionally, recent studies have explored the role of synchronization in the security of quantum key distribution (QKD) protocols [51–53]. It is foreseeable that a better understanding of quantum synchronization could help improve security against specialist attacks that exploit the dependence of several QKD schemes on synchronization during calibration. Despite the intense recent study and the promise it holds, a general theory of quantum synchronization has been lacking until now.

An important theoretical concept in physics is that of symmetries and algebraic structures. Using algebraic methods, it is often possible to analytically understand the dynamics of systems without providing full solutions. This is vitally important because obtaining full solutions is, in general, impossible for the many-body and strongly correlated systems of interest. In this paper, we develop an algebraic theory for quantum synchronization in such systems. This extends the algebraic approach of dynamical symmetries [54] which has been successfully used for studying the emergence of long-time dynamics in various quantum many-body systems, such as time crystals [55–86], quantum scarred models [87–107], Stark many-body localization [108], and models with quantum many-body attractors [109]. Our work provides necessary and sufficient conditions for quantum synchronization to occur in terms of a compact set of algebraic criteria that may be checked based on the underlying symmetry structure of the system. We also rigorously demonstrate that if synchronization occurs, it must be due to the presence of quantum coherence. We fully solve these dynamics in terms of an elegant algebraic framework. These results provide a general theory for studying quantum synchronization and characterize the phenomenon. Through this, we have provided a framework for future efforts in the field.

In the rest of the introduction, we give a general discussion and brief history of the study of synchronization before focusing more closely on the details of quantum synchronization. We finally highlight the key results of this article.

## 1.1 An Overview of Synchronization

### 1.1.1 General Synchronization

Before presenting our theory of quantum synchronization, it is instructive to first discuss synchronization in general, with some historical background, to demonstrate the numerous subtleties surrounding the concept and make it clear what we will be referring to as synchronization.

The term synchronization, or more specifically synchronous, has its roots in the ancient Greek words 'syn-' meaning "together with" and 'kronous' meaning "time". Thus events are described as synchronous if they occur at the same time. The scientific study of synchronous (or synchronized) dynamical systems dates back to 1673 when Huygens studied the motions of two weakly coupled pendula [110]. He observed that pendula clocks hanging from the same bar had matched their frequency and phase exactly. Following this observation, he explored both synchronization and anti-synchronization where the pendula with approximately equal (or opposite for anti-synchronization) initial conditions would, over time, synchronize so that their motions were identical (resp. opposite). Since Huygens, synchronization has been a topic of significant interest in an ever-expanding range of scientific and technological fields. From a theoretical point of view, synchronization is a central issue in the study of dynamical and chaotic systems and is currently an area of very active research.

Broadly speaking, studies into classical synchronization of multiple, often chaotic, systems can be split into synchronization within driven and autonomous systems. Throughout the remainder of this work, we will consider exclusively synchronization within autonomous systems, often called spontaneous synchronization, as this is the form of synchronization most often studied in quantum systems and is where our theory is applicable. For more details regarding synchronization in driven systems, especially the most commonly studied master-slave systems, we direct the reader to [111–113]. We should also mention the related topic of Synergistics, first introduced by Haken to initially study lasers and fluid instabilities [114–116]. Synergetics studies how circular causality between microscopic systems and macroscopic order parameters can lead to self-organization within open systems that have been driven far from equilibrium [117, 118]. In the years since its inception, the theory has been applied to a wide range of disciplines such as studying human ECG activity, and machine learning [119].

We now focus on spontaneous[1] synchronization, which can be further divided into two main classes, which intuitively capture how the two subsystems are related.

(i) Identical/Complete synchronization

*For a system comprising of identical coupled subsystems. Corresponding variables in each of the subsystems become equal as the subsystems evolve in time.*

(ii) Phase synchronization

*For a system comprising of non-identical coupled subsystems. The phase differences between given variables in the different subsystems lock while the amplitudes remain uncorrelated.*

---

[1]From now on, all synchronization will be spontaneous synchronization in undriven systems.

One important additional constraint, which applies to both classes is that synchronization must be robust to small perturbations in the initial conditions. This is important to exclude many cases of trivially synchronized systems that arise simply through the fine-tuning of initial conditions.

In many ways, identical synchronization is the most fundamental and is what Huygens originally studied with his coupled pendula. It is also what many people conjure to mind when thinking about synchronization. As expected, it has been extensively studied in the past, both theoretically [8, 120–124], and for its applications [125–128]. Much of this progress, which has been primarily focused on controlling synchronization within chaotic systems, has built upon the seminal work of Pecora and Carroll [8] who formulated criteria based on the signs of Lyapunov exponents. We also note that identical synchronization can be extended to non-identical subsystems by choosing appropriate, often de-dimensionalized, variables or coordinates for the different subsystems.

A useful order parameter for identical synchronization is the Pearson indicator [129] defined for two time-dependent signals, $f(t), g(t)$ as

$$C_{f,g}(t, \Delta t) = \frac{\overline{\delta f \, \delta g}}{\sqrt{\overline{\delta f^2} \, \overline{\delta g^2}}}, \qquad (1)$$

where $\Delta t$ is some fixed parameter and

$$\overline{X} = \frac{1}{\Delta t} \int_t^{t+\Delta t} X(t) dt, \quad \delta X = X - \overline{X}. \qquad (2)$$

This measures the correlation between the two variables in an intuitive way and has been widely applied to classical [7, 130] and in some cases quantum [36, 42, 131, 132] synchronization. We can see that $C_{f,g}$ takes the value $+1$ for 'perfect' synchronization and $-1$ for 'perfect' anti-synchronization. Notably the Pearson indicator is effective when the signals $f$ and $g$ are not periodic. While synchronization is often understood through periodic systems, such as pendula etc, this is not strictly necessary for identical synchronization. The notion of two variables that become equal over the period of evolution could just as well apply to two particles following unbound trajectories as to two particles that remain within some finite region. We can also easily present examples of quasi-periodic behaviour, that is a superposition of Fourier modes with non-commensurate frequencies. In practice, most systems that we are studying exhibit periodic or at least quasi-periodic behaviour.

In contrast to identical synchronization, phase synchronization generally requires periodic motion to define a relative phase difference between the two subsystems meaningfully. However, extensions can be considered with simply a time delay rather than an actual phase difference. This class of synchronization is weaker than identical synchronization, even under the natural extension to non-identical subsystems, as it requires no direct correlation between the variables describing the subsystems. Following work on cryptography using chaotic maps [133], phase synchronization was applied as a method for secure communication [134, 135]. Phase synchronization has also been studied in the quantum setting through the use of Arnold tongues, and Hussimi-Q functions [37, 129]. These methods are usually aimed at probing the response of the system to driving rather than mutual synchronization between subsystems. We should also mention the interesting related case of amplitude-envelope synchronization in chaotic systems [136] where there is no correlation between the amplitude and phases of the motion within the two systems, but instead, they both develop periodic envelopes at a matched frequency. In the remainder of our work, we will focus on studying identical synchronization between subsystems of an extended quantum system, although many of the results presented are just as applicable to phase synchronization.

Before discussing quantum synchronization in more detail, we make a short remark about the language used in the rest of this article. We will use the term synchronized to refer to any pair of time-dependent signals that are equal or sufficiently similar, usually after some transient time. The term synchronization will refer to the non-trivial process of two subsystems becoming synchronized in a way that is stable to perturbations in the initial conditions. In systems with multiple observable quantities within each subsystem, we will say the subsystems are completely synchronized if all observable quantities are synchronized, and at least one is non-constant. Consequently, we will use identical synchronization for the identical/complete synchronization discussed above to avoid confusion. These definitions will be formalized in Section 2.

### 1.1.2 Quantum Synchronization

Let us now focus more closely on synchronization in many-body quantum systems.

Synchronization is usually studied in open quantum systems. This is because, in closed systems, a generic initial state will excite a large number of eigenfrequencies with random phase differences, which will oscillate indefinitely and lead to observables that behave like noise. Only in sufficiently large systems does the Eigenstate Thermalisation Hypothesis [137] predict that these oscillations will rapidly dephase, causing observables to become effectively stationary. As a result, previous studies have focused on the open quantum system regime, where interactions with the environment cause decoherence and decay within the density operator describing the state, leaving only a handful of long-lived oscillating modes. These systems have previously been studied in a case-by-case manner, with various measures of quantum synchronization having been introduced. These measures have been based primarily on phase-locking of correlations, or Husimi Q-functions [36,37]. However, such measures are often unscalable to many-body problems.

The generic set-up we will consider is an extended system made up of arbitrarily, but finitely, many individual sites that can interact with each other and with an external bath that we often call the environment. This scenario is depicted in Fig 1. The restriction of having only finitely many sites is reasonable because if two subsystems are infinitely far apart, then locality will prevent them from ever being causally connected, thus rendering synchronization impossible. Alternatively, sites infinitely far apart could be redefined as belonging to the environment and thus taken into account in that manner. For our purposes, these individual sites will have finite-dimensional Hilbert spaces and thus have no well defined semi-classical limit. Importantly, these sites will also have identical local Hilbert space dimensions. We will then consider synchronizing the signals of observables measured on different sites or groups of sites of the system. Following our discussion above regarding synchronization in general, we will classify whether the sites are synchronized based on the dynamics of expectation values of these observables on each site over time. In this way, we have captured the fundamental essence of synchronization as discussed above intuitively within the quantum regime.

## 1.2 Overview of the Paper, Summary of Key Results and Relation to Previous Work

In this final part of the introduction, we will outline the structure of the rest of the article and draw attention to the key results which we present.

In Sec. 2 we will introduce the Lindblad formalism as the most general type of smooth Markovian quantum evolution. This formalism is particularly useful since we recover an appealing and intuitive interpretation of the evolution equation in the limit of weak system bath interactions and under assumptions that the bath is Markovian. This section will also formalize our earlier discussion and explicitly define what we mean by synchronization in a quantum

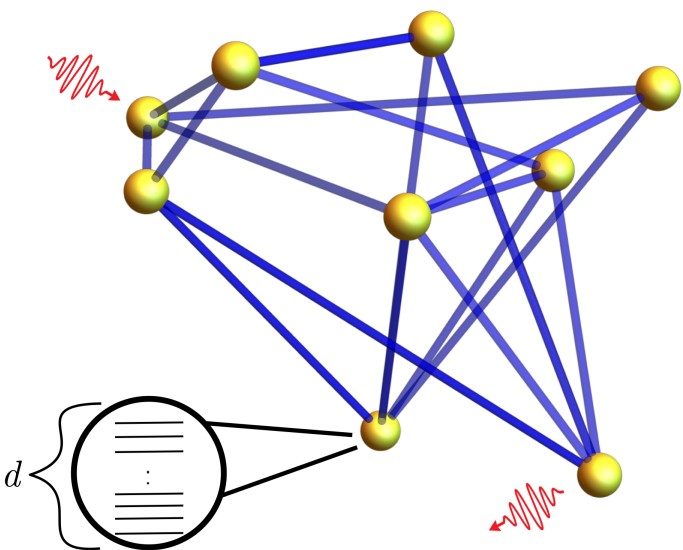

Figure 1: An arbitrary interacting quantum many-body system of $N$ sites (illustrated as yellow spheres) which may interact with each other along the blue bonds. The sites also interact with the background environment, illustrated by the red arrows. The sites each have a finite local Hilbert space, illustrated as dimension $d$ on one of the sites (i.e. each site has exactly $d$ levels). This is the general system of interest we will be focusing on. The goal will be to synchronize observables between different sites in the system. Crucially, the sites that are to be synchronized do not have $\hbar \to 0$ or equivalent large parameter (semi-classical) limits. Rather we make no assumptions on the size of the local Hilbert space.

system. This will be the natural extension of the ideas we presented above and will distinguish between stable and meta-stable synchronization, which may further be robust to our choice of initial conditions or measurements.

In order to develop an algebraic theory of quantum synchronization, we will in Sec. 3.1 first develop a complete algebraic theory of the purely imaginary eigenvalues of quantum Liouvillians and their corresponding *oscillating eigenmodes* that we relate to limit cycles. This theory will be closely related to dynamical symmetries [54], thus demonstrating their importance for understanding long-time non-stationarity in open quantum systems. Previous works on Liouvillian eigenvalues have been based on finding the support of the stationary subspaces or diagonalizing the Hamiltonian [138–141]. However, in the case of many-body systems, both of these calculations are generally impossible except in very specific cases. Although in some cases analytic tools can be used to find non-equilibrium steady states, it remains an open problem to efficiently find the support of a generic stationary state [142]. Therefore, our criteria, based purely on the symmetry structures within the Liouvillian, are easier to work with for a many-body quantum system and can further be used to find systems that present persistent oscillations and limit cycles, an absence of relaxation and synchronization. Additionally, being necessary, our conditions can also be used to prove the absence of such phenomena. The results of this section, which completely characterize all possible purely imaginary eigenvalues of quantum Liouvillians, are far more wide-reaching than just quantum synchronization. In particular, they are also directly relevant for dissipative time crystals [57] and, more generally, any study which is concerned with long-time non-stationarity in an open system.

In Sec. 3.2 we will apply the above theory to stable synchronization, i.e. that exists for infinitely-long times. We will again show that symmetries can be used to provide conditions under which synchronization can occur. These results will highlight the importance of uni-

tal evolutions, i.e. those which preserve the identity operator, for quantum synchronization. Further, within the Lindblad formalism, these unital evolutions are easy to construct by considering dephasing or independent particle loss and gain. This can also explain why several previously studied cases of quantum synchronization are in systems where the evolution is indeed unital.

Having considered infinitely-long lived synchronization, we then proceed in Sec. 4 to provide a perturbative analysis of purely imaginary eigenvalues, again with a focus on the consequences for quantum synchronization. We classify the resulting cases into ultra-low frequency, quantum Zeno and dynamical metastable synchronization. As we show, ultra-low frequency metastable synchronization is rather unsatisfactory for experimental purposes since the time period of the resulting oscillations is exceptionally long. We further prove that dynamical metastable synchronization has the longest lifetime and has oscillations that occur on relevant experimental timescales. Thus we conclude this is the most desirable form of metastable synchronization. Arguably, we have at this point covered all possible cases of quantum synchronization.

In Sec. 5 by taking the converse approach, we extend our algebraic theory so that it can be used to prove the *absence* of quantum synchronization in a given system. This is useful for easily identifying which systems should not be considered when looking for candidates for quantum synchronization.

Having completely characterized quantum synchronization through an analysis of Liouvillian eigenvalues, in Sec . 6 we apply our theory in a simple example where we show how two qubits may be *anti-synchronized* complementing the recent claim that qubits cannot synchronize [143]. We further use our results to demonstrate that the regularly studied example of quantum synchronization in the Fermi-Hubbard model with spin-agnostic heating [36] is, in fact, one specific example of a wide range of systems that can be expected to exhibit quantum synchronization. We will relate this discussion to more experimentally relevant set-ups with a less strict symmetry structure. Our theory then allows us to give simple predictions to explain experimental results previously found by [144].

Finally, we present a conclusion. Detailed proofs of our results together with additional analysis of existing examples of quantum synchronization are presented in the appendices.

We note that criteria for purely imaginary eigenvalues of quantum Liouvillians have been given in other papers, e.g. [138,139]. However, they require either diagonalizing the Hamiltonian or the stationary state to check and use. Both of these steps are prohibitively difficult for many systems once their size becomes sufficiently large. Moreover, they do not allow for the exact form of the corresponding eigenmodes (quantum limit cycles). Our work will rely on dynamical symmetries and immediately gives the form of the corresponding eigenmodes if the symmetries are known. Dynamical symmetries, first introduced in the context of quantum Liouvillians [54], have been applied before to studying certain cases of quantum synchronization [36]. In these works, dynamical symmetries were shown to be sufficient for the existence of purely imaginary eigenvalues, and synchronization enabled by these symmetries was studied. In this work, we will give both necessary and sufficient algebraic conditions for the existence of purely imaginary eigenvalues which can then be directly related to sufficient criteria for quantum synchronization. Moreover, we show that the previously identified compact sufficient condition is also necessary for quantum synchronization in *unital* quantum Liouvillians, which for example, includes those that have thermal stationary states.

## 2  Definitions and Methods

In this section, we will first present the natural definitions of synchronization in quantum systems, as follow from our previous discussion and then introduce the Lindblad Master equation framework as the most general description of the evolution of a quantum state.

### 2.1  Definitions of Quantum Synchronization

To recap our discussion above, for two systems to be identically synchronized, we require that their matching motion be non-stationary and long-lasting. For an extended quantum system, we will interpret the 'motion' via the behaviour of some local observable, $O_j$, which is measured in the same basis on every site. Note that we will use subscripts to denote the subspaces/sites on which the operator acts non-trivially i.e. $O_j = \mathbb{1}_{1...j-1} \otimes O_j \otimes \mathbb{1}_{j+1...N}$ with $N$ being the number of subsystems/sites. We will also consider only the strictest notion of identically synchronized signals whereby after synchronization has occurred, the two synchronized signals, $\langle O_j(t)\rangle$ & $\langle O_k(t)\rangle$, are identical and do not differ by any overall phase, scale factor, or constant. This is stricter that the definition provided by considering the Pearson indicator from Eq. (1). The results we present can be suitably adapted to consider these alternative cases, in particular phase synchronization. We will indicate in Sec. 3.4 how alternate considerations can be made, but as the technical discussions do not provide any additional insight or understanding, we will work largely with these very strict definitions given below.

In line with previous works, we consider stable and metastable synchronization to be where the signals remain synchronized and non-stationary for infinitely long or finitely long times, respectively. In the case of metastable synchronization, we require some perturbative parameter within the system that controls the lifetime of the synchronized behaviour. Finally, we must allow some initial transient time period during which the process of synchronization can take place. These considerations lead naturally to the following definitions.

Note that equality in these definitions, and in the remainder of this work, is understood as equality up to terms which are exponentially small in time and for brevity, we do not continuously write "$+\mathcal{O}(e^{-\gamma t})$". These terms will always be present for finite-dimensional systems but are negligible on the longer time-scales we will be concerned with.

**Definition 1** (Stably synchronized)**.** We say that the subsystems $j$ and $k$ are *stably synchronized* in the observable $O$ if for some initial state and after some transient time period $\tau$ we have $\langle O_j(t)\rangle = \langle O_k(t)\rangle$ for all $t \geq \tau$. Further we require that $\langle O_j(t)\rangle$ does not become constant, i.e $\partial_t \langle O_j(t)\rangle \not\equiv 0$.

**Definition 2** (Metastably[2] synchronized)**.** We say that the subsystems $j$ and $k$ are *metastably synchronized* in the observable $O$ if for some initial state during the interval $t \in [\tau, T]$ where $T \gg \tau$ we have $\langle O_j(t)\rangle = \langle O_k(t)\rangle$ and again $\partial_t \langle O_j(t)\rangle \not\equiv 0$. Further, the cut-off time $T$ must be controllable by some perturbative parameter in the system

These definitions do not require that the system has an internal mechanism that synchronizes the two sites. Since we simply require the existence of some initial state for which the observables are synchronized, this initial state can be finely tuned so that, in fact, the observables on the two subsystems are initially equal, and their evolutions are identical. To characterize synchronization that is robust to variation in initial conditions, we make the further definition.

---

[2]Metastable synchronisation of this form has also been previously referred to as transient synchronisation [44, 145].

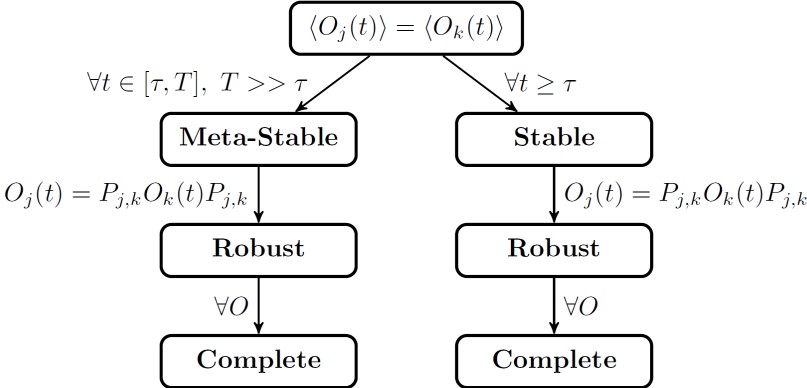

Figure 2: Visualization of the different definitions of synchronization. If we have equality in the expectation value of some non-stationary observable on different sites for some initial state, then we classify the signals as either stably or metastably synchronized. If, in fact, we have equality on the operator level, then they are robustly synchronized. Finally, if this holds for all non-stationary operators, then we say the subsystems are completely synchronized.

**Definition 3** (Robustly synchronized). If the subsystems $j$ and $k$ are stably or metastably synchronized in the observable $O$, this synchronization is *robust* if $O_j(t) = P_{j,k}O_k(t)P_{j,k}$ and $\partial_t O_j(t) \not\equiv 0$ on the operator level (i.e. in the Heisenberg picture). Here $P_{j,k}$ is a permutation operator exchanging subsystems $j$ and $k$. As with the definitions of stable and meta-stable synchronization, these requirements must hold for the same respective time periods.

This definition ensures that after the transient dynamics have decayed, regardless of the system's initial state, the observable on subsystems $j$ and $k$ will be equal and for a large class of initial states will be non-stationary. Robustly synchronized systems are of the most significant interest since these are the ones where some internal mechanism causes the synchronization process. However, as demonstrated in the appendices, several previously studied examples of quantum synchronization are, in fact, not robust and require fine-tuning of the initial state. This highlights the importance of making these considerations and definitions when studying quantum synchronization. We also remark that we have restricted ourselves to the case of identical subsystems so that robustness can be defined in this way. If one were to study synchronization between a spin-1/2 and a spin-1, for example, it is unclear how robustness could be defined in a similarly straightforward manner.

Notice that we have defined synchronization with respect to some observable. For most applications of synchronized motion, it is sufficient for just one observable to be synchronized. We can, however, say that the two subsystems are *completely* synchronized if they are robustly synchronized, whether stably or metastably, for *all* non-stationary observables [16].

These definitions are visualized in Fig. 2 and characterize the possible cases of identical synchronization within a quantum system. As earlier remarked, the results we present can be straightforwardly adapted to also consider phase synchronization provided the long-lived oscillations are periodic. Moreover, we can easily extend our analysis to other measures of synchronization [146]. For instance, we may define limit cycle dynamics over a phase space composed of $\langle O_k(t) \rangle$ and another observable $\langle Q_k(t) \rangle$ that obeys $[Q_k, O_k] \neq 0$.

Having now made explicit what we mean when referring to quantum synchronization and introduced some of the notation, we can begin to study these systems in more detail.

## 2.2 Lindblad Formalism

We describe the quantum state by a time dependent density operator, $\rho(t)$, acting on the Hilbert space of the system, $\mathcal{H}$. The most general quantum evolution is described by a completely positive, trace preserving channel $\hat{\mathcal{T}}_t$ so that after a time $t$ the state has evolved via

$$\rho(t) = \hat{\mathcal{T}}_t[\rho(0)]. \tag{3}$$

It is known [147] that any smooth, time-homogeneous, completely positive, trace-preserving quantum channel $\hat{\mathcal{T}}_t[\rho]$ which obeys the natural semi-group property

$$\hat{\mathcal{T}}_t \cdot \hat{\mathcal{T}}_s = \hat{\mathcal{T}}_{t+s}, \tag{4}$$

can be expressed as

$$\hat{\mathcal{T}}_t[\rho] = e^{t\hat{\mathcal{L}}}[\rho], \tag{5}$$

and thus

$$\frac{d}{dt}\rho = \hat{\mathcal{L}}[\rho]. \tag{6}$$

Here $\hat{\mathcal{L}}$ is a quantum Liouvillian of the form

$$\hat{\mathcal{L}}[\rho] = -\mathrm{i}[H,\rho] + \sum_\mu \left( 2L_\mu \rho L_\mu^\dagger - \{L_\mu^\dagger L_\mu, \rho\} \right), \tag{7}$$

for some Hermitian operator $H$. If we are considering a system which weakly interacts with a Markovian environment then equations (6-7) are usually referred to as the Lindblad master equation and we can interpret the operator $H$ as the system's Hamiltonian and the $L_\mu$ operators, now called Lindblad jump operators, model the influence of the environment's noise on our system. Such operators may, for example, be particle creation/annihilation operators to describe random particle gain/loss, or number operators to describe dephasing [148]. By formulating our theory of quantum synchronization with respect to evolutions described by Eqs. (6-7) the resulting theory is as general as possible and can be intuitively understood in the weak coupling, Markovian limit. However, evolution of the quantum state according to Eqn. (7) is not restricted exclusively to the weak coupling limit, although at strong coupling the jump operators lose their intuitive interpretation. By formulating our theory within the Lindblad formalism we are able to capture the widest class of possible quantum evolutions which are relevant to undriven systems which interact with an environment.

We remark that it is possible to also include some non-Markovian effects in this formalism by considering an enlarged Hilbert space containing part of the environment, as we outline further in Appdx K. Although this is usually unpractical for large many-body systems, the example in Sec 6.1 will in effect use this idea when anti-synchronizing two spin-1/2s. With this one exception, however, our work considers exclusively Markovian dynamics.

A formal solution to the Lindblad equation can be obtained by finding the eigensystem of the Liouvillian. This is defined as the set $\{\rho_k, \sigma_k, \lambda_k\}$, where $\lambda_k$ are the eigenvalues of $\hat{\mathcal{L}}$ and $\rho_k, \sigma_k$ are the corresponding right and left eigenstates respectively. The eigensystem obeys the relations,

$$\hat{\mathcal{L}}[\rho_k] = \lambda_k \rho_k, \ \hat{\mathcal{L}}^\dagger[\sigma_k] = \lambda_k^* \sigma_k,$$
$$\langle\langle \sigma_k | \rho_{k'} \rangle\rangle = \delta_{k,k'}, \tag{8}$$

where $\langle\langle \sigma | \rho \rangle\rangle = \mathrm{Tr}(\sigma^\dagger \rho)$ is the Hilbert-Schmidt inner product. Note that for operators $\hat{\mathcal{L}}$ which generate a CPTP map and thus describe physical quantum evolutions, the eigenvalues, $\lambda_k$, can lie only in the left half of the complex plane with $\mathrm{Re}(\lambda_k) \leq 0$. In the familiar way,

assuming diagonalizability of $\hat{\mathcal{L}}$ we can express the evolution of the expectation value of some observable $\hat{O}$, given that the system is initialised in the state $|\rho_0\rangle\rangle$ as

$$\langle \hat{O} \rangle(t) = \sum_k e^{t\lambda_k} \langle\langle \hat{O}|\rho_k\rangle\rangle \langle\langle \sigma_k|\rho_0\rangle\rangle. \tag{9}$$

We now see that in order to achieve long-lived oscillations in some observable, there must exist eigenvalues $\lambda_k$ with non-zero imaginary part and vanishingly small real part. Note that in general, $\hat{\mathcal{L}}$ may not be diagonalizable, but that its restriction to the asymptotic subspace always is (Appendix F). Therefore in the long-time limit, we can assume that (9) is the evolution equation for all cases.

The above observation motivates the next section, where we present algebraic results for the existence of purely imaginary eigenvalues in quantum Liouvillians. Following these results, we shall use them to provide spectral and symmetry-based conditions for quantum synchronization as described by our previous definitions. We will then demonstrate these conditions in a variety of paradigmatic and insightful examples before concluding.

# 3 Imaginary Eigenvalues of Quantum Liouvillians and Stable Synchronization

In this section, we present a series of algebraic results on the existence of purely imaginary eigenvalues, give the structure of the corresponding eigenmodes, and relate them to stable quantum synchronization. These are important not only for quantum synchronization, but also for dissipative time crystals [54, 57, 67, 70–72, 149–155] and more generally any study of non-stationarity in open quantum systems. All proofs for the following results can be found in the Appendices. As mentioned previously, we will focus on systems with an arbitrarily large but strictly finite number of local levels (i.e. a finite local Hilbert space).

## 3.1 Necessary and Sufficient Conditions for Purely Imaginary Eigenvalues and the Structure of the Oscillating Coherent Eigenmodes

Our first theorem completely characterizes all purely imaginary eigenvalues of Liouvillian super-operators in terms of a unitary operator $A$ and a proper non-equilibrium steady state (NESS) of the system $\rho_\infty$ which obeys $\hat{\mathcal{L}}[\rho_\infty] = 0$. Note that by 'proper', we mean that $\rho_\infty$ is a density operator and can thus represent an actual quantum state of the system.

**Theorem 1.** *The following condition is necessary and sufficient for the existence of an eigenstate $\rho$ with purely imaginary eigenvalue $i\lambda$, $\hat{\mathcal{L}}\rho = i\lambda\rho$, $\lambda \in \mathbb{R}$.*

*We have $\rho = A\rho_\infty$, where $\rho_\infty$ is a NESS and $A$ is a unitary operator which obey,*

$$[L_\mu, A]\rho_\infty = 0, \tag{10}$$

$$\left( -i[H, A] - \sum_\mu [L_\mu^\dagger, A]L_\mu \right)\rho_\infty = i\lambda A\rho_\infty, \; \lambda \in \mathbb{R}. \tag{11}$$

The necessity direction of the proof in Appx. A is a consequence taking the polar decomposition of $\rho = AR$ and then showing that the unitary part, $A$, must satisfy the stated conditions, while the positive-semidefinite part, $R$, satisfies $\hat{\mathcal{L}}[R] = 0$ and is thus a proper NESS. The sufficiency direction can be obtained by rearranging Eqs. (10) and (11) and does not require that $A$ be unitary. Note that we have solved for the eigenmode in terms of $A$ and $\rho_\infty$. We also

emphasize that the necessity of the conditions in Th. 1 does not hold for infinite-dimensional systems (e.g. bosons), but the sufficiency does. From this theorem, we can also obtain alternative necessary conditions for the existence of a purely imaginary eigenvalue.

**Corollary 1.** *The Liouvillian super-operator, $\hat{\mathcal{L}}$, admits a purely imaginary eigenvalue, $i\lambda$ only if there exists some unitary operator, $A$, satisfying*

$$-i\rho_\infty A^\dagger[H,A]\rho_\infty = i\lambda\rho_\infty^2, \tag{12}$$

$$\rho_\infty A^\dagger[L_\mu^\dagger,A]L_\mu\rho_\infty = 0, \;\; \forall\mu, \tag{13}$$

$$\rho_\infty[L_\mu^\dagger,A^\dagger]L_\mu\rho_\infty = 0, \;\; \forall\mu. \tag{14}$$

*In which case the eigenstate is given by $\rho = A\rho_\infty$.*

In Appx. B this is shown by rearranging the conditions in Eqs. (10) and (11).

We next make the connection between these results and strong dynamical symmetries [54] which have previously been shown as sufficient for the existence of purely imaginary eigenvalues. Using Thm. 1, we can, under certain conditions, strengthen the results of [54] and demonstrate that strong dynamical symmetries are both necessary and sufficient.

**Theorem 2.** *When there exists a faithful (i.e. full-rank/ invertible) stationary state, $\tilde{\rho}_\infty$, $\rho$ is an eigenstate with purely imaginary eigenvalue if and only if $\rho$ can be expressed as,*

$$\rho = \rho_{nm} = A^n \rho_\infty (A^\dagger)^m, \tag{15}$$

*where $A$ is a strong dynamical symmetry obeying*

$$[H,A] = \omega A \tag{16}$$

$$[L_\mu,A] = [L_\mu^\dagger,A] = 0, \;\; \forall\mu, \tag{17}$$

*and $\rho_\infty$ is some NESS, not necessarily $\tilde{\rho}_\infty$. Moreover the eigenvalue takes the form*

$$\lambda = -i\omega(n-m). \tag{18}$$

*Furthemore, the corresponding left eigenstates, with $\hat{\mathcal{L}}^\dagger \sigma_{mn} = i\omega(n-m)\sigma_{mn}$, are given by $\sigma_{mn} = (A')^m \sigma_0 ((A')^\dagger)^n$ where $A'$ is also a strong dynamical symmetry and $\sigma_0 = \mathbb{1}$.*

The proof of the necessity direction in Appx. C makes extensive use of the results and proofs from [139, 156, 157] regarding the asymptotic subspace of the Liovillian, $As(\mathcal{H})$ and the projector, $P$, into the corresponding non-decaying part of the Hilbert space, $\mathcal{H}$. Importantly, when a full rank stationary state exists, this projector must be the identity operator, $P = \mathbb{1}$. Compared with Thm. 1 there is now no ambiguity about the unitarity of $A$. The simple manipulation

$$[H,A] = \omega A$$
$$A^{-1}HA - H = \omega\mathbb{1} \quad \text{Assuming } A \text{ is invertible} \tag{19}$$
$$0 = \omega\text{Tr}(\mathbb{1}) \quad \text{Taking trace}$$

shows that $A$ must *not* be invertible and thus cannot be unitary when $\omega \neq 0$. Additionally, taking the trace of Eq. (16) shows $A$ must be traceless for $\omega \neq 0$. We also remark that the requirement of a faithful stationary state, i.e that $\rho_\infty$ has full rank, is immediately satisfied if the Liouvillian is unital, defined as $\hat{\mathcal{L}}[\mathbb{1}] = 0$ where $\mathbb{1}$ is the identity operator. This simplifies to $\sum_\mu[L_\mu, L_\mu^\dagger] = 0$ which in particular is true for pure dephasing where $L_\mu = L_\mu^\dagger$. We will give further discussion of unital maps and their relevance for synchronization later. Importantly, the quantum limit cycle eigenmodes $\rho_{nm}$ are, by construction, off-diagonal in the basis of $\rho_\infty$ indicating that they are quantum coherent (cf. also quantum phase synchronization [158]).

## 3.2 Stable Quantum Synchronization

The criteria for persistent oscillations we gave in the previous subsection are necessary for stable quantum synchronization because the sites in a system cannot lock persistently into phase, frequency and amplitude if the frequency is trivially 0. The sufficient criteria depend on the precise definition of synchronization, as we will now examine. We will use the terminology introduced in Sec. 2.1. Recall that the crucial feature of quantum synchronization is that the various parts of the subsystem lock into the same phase, frequency and amplitude. Let $P_{j,k}$ be an operator exchanging subsystems $j$ and $k$ and let $\hat{\mathcal{P}}_{j,k}(x) := P_{j,k} x P_{j,k}$. We note that $P_{j,k}^2 = \mathbb{1}$. We can then prove the following corollary.

**Corollary 2.** *Fulfilling all the following conditions is sufficient for robust and stable synchronization between subsystems $j$ and $k$ with respect to the local operator $O$:*

- *The operator $P_{j,k}$ exchanging $j$ and $k$ is a weak symmetry [159] of the quantum Liouvillian $\hat{\mathcal{L}}$ (i.e. $[\hat{\mathcal{L}}, \hat{\mathcal{P}}_{j,k}] = 0$)*

- *There exists at least one $A$ fulfilling the conditions of Th. 1*

- *For at least one operator $A$ fulfilling the conditions of Th. 1 and the corresponding $\rho_\infty$ we have $\mathrm{tr}[O_j A \rho_\infty] \neq 0$*

- *All $A$ fulfilling the conditions of Th. 1 also satisfy $[P_{j,k}, A] = 0$*

The proof detailed in Appx. D follows because under the assumption that the exchange operator, $P_{j,k}$, is a weak symmetry we find that all NESSs, $\rho_\infty$, with $\hat{\mathcal{L}}[\rho_\infty] = 0$ commute with $P_{j,k}$. Thus when we write the general evolution in the long time limit we have

$$
\begin{aligned}
\lim_{t\to\infty} \langle O_j(t) \rangle &= \mathrm{tr}\left[ O_j \sum_n c_n A_n \rho_{\infty,n} e^{i\lambda_n t} \right] \\
&= \mathrm{tr}\left[ O_j P_{j,k}^2 \sum_n c_n A_n \rho_{\infty,n} e^{i\lambda_n t} \right] \\
&= \mathrm{tr}\left[ P_{j,k} O_j P_{j,k} \sum_n c_n A_n \rho_{\infty,n} e^{i\lambda_n t} \right] \\
&= \mathrm{tr}\left[ O_k \sum_n c_n A_n \rho_{\infty,n} e^{i\lambda_n t} \right] \\
&= \lim_{t\to\infty} \langle O_k(t) \rangle,
\end{aligned}
\tag{20}
$$

where: $c_n = \langle\langle \sigma_n | \rho(0) \rangle\rangle$ for an arbitrary initial state $\rho(0)$, the operators $A_n$ all satisfying the conditions of Th. 1 and $\hat{\mathcal{L}}\rho_{\infty,n} = 0$.

The most straightforward example of a weak symmetry is when $[H, P_{j,k}] = 0$ and $\hat{\mathcal{P}}_{j,k}$ maps the set of all the Lindblad operators $\{L_\mu\}$ into itself, though more exotic cases are possible (e.g. [160–164]). Thus systems satisfying the conditions of Cor. 2 are, for instance, those for which $P_{j,k}$ is a reflection operator, the Lindblad operators act on each subsystem individually, and the system Hamiltonian is invariant under reflections.

We may further relax these requirements in quite general cases and achieve total synchronization across the system, i.e. synchronization between all pairs of subsystems. In Appx. E we prove the following refinement of Cor. 2.

**Corollary 3.** *The following conditions are sufficient for robust and stable synchronization between subsystems $j$ and $k$ with respect to the local operator $O$:*

- *The Liouvillian, $\hat{\mathcal{L}}$ is unital ($\hat{\mathcal{L}}(\mathbb{1}) = 0$)*

- *There exists at least one operator $A$ fulfilling the conditions of Th. 1*

- *For at least one $A$ fulfilling the conditions of Th. 1 and the corresponding $\rho_\infty$ we have $\mathrm{tr}[O_j A \rho_\infty] \neq 0$*

- *All such $A$ fulfilling the conditions of Th. 1 also satisfy $[P_{j,k}, A] = 0$*

*Furthermore, if all $A$ are translationally invariant, $[A, P_{j,j+1}] = 0, \forall j$, then every subsystem is robustly and stably synchronized with every other subsystem.*

This further illustrates the power and utility of unital maps in achieving total synchronization. Examples are $1D$ models that are reflection symmetric, have only one $A$ operator, and experience dephasing. This includes the cases discussed before in [36, 54].

## 3.3 Multiple Frequencies and Commensurability

Before moving on, we make some brief remarks about the issues of multiple frequencies and commensurability. Firstly, since our theory allows for multiple $A$ operators, each of which needs not correspond to the same imaginary eigenvalue, the theorems above explicitly include the case of multiple frequencies. In general, if the multiple purely imaginary eigenvalues are not commensurate, i.e. are not all integer multiples of some fixed value, then the resulting oscillations will not be periodic. This alone does not preclude quantum synchronization since we have not required that the long-lived dynamics be periodic.

However, if there are "too many" incommensurable purely imaginary eigenvalues, they will generically dephase, leading to observables that are effectively stationary - at least within experimentally accessible and measurable limits. This is similar to eigenstate dephasing, which is often considered the generic mechanics for thermalization in closed systems [137, 165]. Alternatively, the system may display chaotic dynamics. This means that in systems obeying the conditions for synchronization but with a large number of incommensurable purely imaginary eigenvalues, additional analysis must be carried out to determine if the synchronization survives the dephasing process and whether the dynamics become chaotic. Importantly, if all the strong dynamical symmetries or $A$-operators satisfy the conditions for Cors 2 or 3 the dynamics will be synchronized at late times even if the evolution is chaotic or relaxes to stationarity. For a more detailed discussion of such spectral problems from a more mathematical perspective, see [165].

It should also be noted that the situation is more straightforward in unital evolutions, where each purely imaginary eigenvalue can be related to a strong dynamical symmetry, $A$, and thus by Thm. 2 is an integer multiple of some fixed frequency corresponding to $A$. Generically in open quantum systems with sufficiently many Lindblad jump operators, there are very few, if any, dynamical symmetries, and thus any purely imaginary eigenvalues will be integer multiples of a few fixed values.

## 3.4 Extensions to Weaker Definitions of Synchronization

As we have emphasised throughout, our definitions pertain only to the strictest notion of synchronization, where the two signals must be identical. However, if we were to use the Pearson indicator from Eqn. (1), this would instead allow for two signals which differ by an overall additive constant or multiplicative factor. It is immediate from Eqn. (20) how Cor. 2 should

be adapted to give sufficient conditions for this relaxed notion of synchronization. Firstly we no longer need exchange superoperator $\hat{\mathcal{P}}_{j,k}$ to be a weak symmetry and we do not need all $A$-operators to satisfy $[P_{j,k}, A] = 0$. Instead we require that if $\rho_n = A_n \rho_{\infty,n}$ has a non-zero, purely imaginary eigenvalue, then $P_{j,k}\rho_n = \alpha \rho_n P_{j,k}$ for some $\alpha$ which corresponds to the constant multiplicative factor between the two signals. Note that inhomogeneity of the NESSs under exhange, $P_{j,k}\rho_{\infty,n}P_{j,k} \neq \rho_{\infty,n}$ leads to an additive constant between the two signals.

As an extension to this we can also see how alternate modes of synchronisation can occur. For instance if there were only a single $A$ operator and the condition $[P_{j,k}, A] = 0$ were replaced by $P_{j,k}A - e^{i\theta}AP_{j,k} = 0$ then we would have

$$\lim_{t\to\infty} \langle O_j(t) \rangle = \lim_{t\to\infty} \langle O_k(t + \theta/\lambda) \rangle, \tag{21}$$

corresponding to phase synchronisation

The results in this section have considered only the cases of purely imaginary eigenvalues. Thus they correspond to infinitely long-lived synchronized oscillations, i.e. *stable* quantum synchronization. In the next section, we will analyze the behaviour of Liouvillian eigenvalues under perturbations and discuss the relation of these results to metastable quantum synchronization.

## 4 Almost Purely Imaginary Eigenvalues and Metastable Quantum Synchronization

Let us now suppose that our Liouvillian is perturbed analytically so that we may write

$$\hat{\mathcal{L}}'(s) = \hat{\mathcal{L}} + s\hat{\mathcal{L}}_1 + s^2\hat{\mathcal{L}}_2 + \mathcal{O}(s^3). \tag{22}$$

For simplicity and to avoid unwanted technicalities, we will assume that this series expansion has an infinite radius of convergence. This is almost always the case in relevant examples where the Hamiltonian or Lindblad jump operators are generally perturbed only to some finite order.

As explained in Sec. 2.1, metastable quantum synchronization requires eigenvalues with vanishingly small real part. Under a perturbation of the form in equation (22), it is known that the eigenvalues, $\lambda(s)$ vary continuously with $s$ [138]. Therefore in order to obtain vanishingly small real parts, it is clear that we must perturb those eigenvalues already on the imaginary axis. At this point we divide our analysis into two regimes, the first where $\lambda(0) = 0$ and the second where $\lambda(0) = i\omega$, $\omega \in \mathbb{R} \setminus \{0\}$.

### 4.1 Ultra-Low Frequency Metastable Synchronization

In the first regime, we consider the case of perturbing a state with zero eigenvalue at $s = 0$. Since there always exists a Liouvillian eigenstate with zero eigenvalue, this is only possible when the null space of $\hat{\mathcal{L}}$ is degenerate. This regime has been studied in depth by Macieszczak et al. [166]. It was shown that when $\hat{\mathcal{L}}'(s)$ is perturbed away from $s = 0$ the degeneracy in the 0 eigenvalue is lifted in such a way that the eigenvalues are at least twice continuously differentiable,

$$\lambda(s) = i\lambda_1 s + \lambda_2 s^2 + o(s^2), \tag{23}$$

where $\lambda_1 \in \mathbb{R}$ and $\text{Re}(\lambda_2) \leq 0$. This gives rise to periodic oscillations with time period $\sim 1/s$ and lifetime $\sim 1/s^2$.

In many ways, this is somewhat unsatisfactory for the purposes of synchronization in an applicable sense since as we extend the lifetime of our synchronization, the periodic behaviour becomes harder to observe as a consequence of the extended time period. We also remark that generically the order of $s$ only differs by one between the decay rate and the oscillation frequency.

### 4.2 Quantum Zeno Metastable Synchronization

A more relevant variant of the ultra-low frequency case is when,

$$\hat{\mathcal{L}}'[\rho] = -\mathrm{i}[H, \rho] + \gamma \hat{\mathcal{D}}[\rho] + \mathcal{O}\left(\frac{1}{\gamma}\right), \tag{24}$$

for some large $\gamma$. In this case the evolution is dominated by the dissipative term

$$\hat{\mathcal{D}}[\rho] = \sum_{\mu} 2L_{\mu}\rho L_{\mu}^{\dagger} - \{L_{\mu}^{\dagger}L_{\mu}, \rho\}. \tag{25}$$

In this set up *quantum Zeno dynamics* are possible. Physically this can, for example, correspond to experimental set-ups in regimes where it is difficult to suppress the effects of dephaing such as the example considered later in Sec 6.3. For more detailed derivations and in-depth discussions of the quantum Zeno effect, see [167–170]. We now re-scale the full quantum Liouvillian as $\tilde{\hat{\mathcal{L}}} = \frac{1}{\gamma}\hat{\mathcal{L}}'$, and consider $s = 1/\gamma$ as the perturbative parameter. If the 0 eigenvalue is split then as in the above section we have

$$
\begin{aligned}
\tilde{\lambda} &= \mathrm{i}\omega s + \lambda_2 s^2 + o(s^2) \\
&= \mathrm{i}\omega \frac{1}{\gamma} + \lambda_2 \frac{1}{\gamma^2} + o(s^2).
\end{aligned}
\tag{26}
$$

Transforming back to the unscaled Liouvillian we now have

$$\lambda = \mathrm{i}\omega + \lambda_2 \frac{1}{\gamma} + o\left(\frac{1}{\gamma}\right), \tag{27}$$

and thus we see that the oscillations occur on the relevant Hamiltonian time-scales for all $\gamma$. In this case the purely imaginary eigenvalues must come from *stationary phase relations* [138] of the dissiaptive Liouvillian $\hat{\mathcal{D}}$. These are stationary states that fulfil the conditions of Th. 1 with trivial eigenvalue $\omega = 0$. If all the $A$ operators and $\hat{\mathcal{D}}$ fulfil the conditions of Cor. 2 or 3, then robust metastable synchronization occurs with a lifetime $\sim \frac{1}{\gamma}$.

### 4.3 Dynamical Metastable Synchronization

We now analyse the case where $\lambda(0) = \mathrm{i}\omega$, with $\omega \neq 0$, which we label dynamical metastable synchronization. This is physically distinct from the case where $\omega = 0$ as we are now considering perturbations to a system which already exhibits long-lived dynamics. By adapting the analysis of [166], in Appx. F we prove the following theorem.

**Theorem 3.** *For analytic $\hat{\mathcal{L}}'(s)$ with $\lambda(0) = \mathrm{i}\omega$, $\omega \in \mathbb{R}$ we have*

$$\lambda(s) = \mathrm{i}\omega + \mathrm{i}\lambda_1 s + \lambda_2 s^{1+\frac{1}{p}} + o\left(s^{1+\frac{1}{p}}\right), \tag{28}$$

*for some integer $p \geq 1$. We also find that $\lambda_1 \in \mathbb{R}$ and $\lambda_2$ has non-positive real part.*

Crucially, this means that at first order in $s$ the perturbed eigenvalues remain purely imaginary, while the real part, which would contribute to decay, is higher order in $s$.

When considering the synchronization aspects, we first note that the conditions for robust metastable synchronization remain the same as in Cor. 2 and 3 in the leading order, i.e. for $\hat{\mathcal{L}}$ (see Appx. J). For sake of simplicity we assume that $\hat{\mathcal{L}}_1$ is such that it explicitly breaks the exchange symmetry $P_{j,k}$ between the sites we wish to synchronize, i.e. $\hat{\mathcal{P}}_{j,k}\hat{\mathcal{L}}_1\hat{\mathcal{P}}_{j,k} = -\hat{\mathcal{L}}_1$. This is the most relevant and standardly studied case in quantum synchronization. For instance, this case occurs if the subsystems $j, k$ are perfectly tuned to the same frequency in $\hat{\mathcal{L}}$ in the leading order, and we introduce a small detuning (e.g. [36]). In that case, we find the remarkable fact that,

**Corollary 4.** *For anti-symmetric perturbations $\hat{\mathcal{L}}_1$ which explicitly break the exchange symmetry, and thus synchronization, between sites $j$ and $k$, i.e. $\hat{\mathcal{P}}_{j,k}\hat{\mathcal{L}}_1\hat{\mathcal{P}}_{j,k} = -\hat{\mathcal{L}}_1$, the frequency of synchronization $\omega$ is stable to next-to-leading order in $s$, i.e. $\lambda_1 = 0$.*

This counter-intuitive result, proven in Appx. G implies that the frequency at which synchronized observables oscillate is more stable to perturbations that disturb the synchronization through explicitly breaking the exchange symmetry than those which preserve this symmetry.

The preceding theorem may be strengthened in the case where $\lambda(0)$ is non-degenerate or where the perturbation of $\hat{\mathcal{L}}'(s)$ corresponds to at least a second-order perturbation of the jump operators as follows

**Corollary 5.** *For a Liouvillian of the form in Eq. (7) with*

$$
\begin{aligned}
H(s) &= H^{(0)} + sH^{(1)} + \mathcal{O}(s^2), \\
L_\mu(s) &= L_\mu^{(0)} + \mathcal{O}(s^2), \ \forall \mu,
\end{aligned}
\tag{29}
$$

*the exponent in Theorem 3 has $p = 1$ so that the eigenvalues are twice continuously differentiable in $s$.*

The proof can be found in Appx. H and follows from noticing that $\hat{\mathcal{L}}^{(1)} = -\mathrm{i}[H^{(1)}, (\cdot)]$ generates unitary dynamics. This result should be contrasted with the ultra-low frequency case. Under these type of perturbations we see that time period of the oscillations does not grow as $s \to 0$ and instead remains $\mathcal{O}(1)$. We also observe that there are two orders of $s$ between the decay rate and oscillation frequency, demonstrating that these are more stable and thus easier to observe experimentally and more relevant for utilisation than the ultra-low frequency case.

## 4.4 Analogy With Classical Synchronization

Before continuing to the examples, we pause to discuss the relationship between quantum synchronization in our sense and classical synchronization. Although there is no well-defined classical limit for the quantum systems and the microscopic observables we study, certain analogies can be drawn.

Firstly, classical synchronization, by definition, requires stable limit cycles [171]. More specifically, suppose that a finite perturbation in the neighborhood of a limit cycle $\langle O(t)\rangle$ leads to a new trajectory $\langle \tilde{O}(t)\rangle$. The limit cycle is exponentially stable if there exists a finite $a > 0$ such that $|\langle \tilde{O}(t)\rangle - \langle O(t)\rangle| \lesssim e^{-at}$. In our case, provided the conditions of Th. 2 hold, any perturbation of the state $\rho(t) + \delta\rho$ that does not change the value of the strong dynamical symmetry, i.e. $\mathrm{Tr}(\delta\rho A) = 0$ renders the limit cycle of $\langle O(t)\rangle$ exponentially stable provided that the Jordan normal form is trivial. Otherwise, it is stable. This follows directly by linearity from Eqs.(8), (9) and Th. 2. We have previously used this fact implicitly to guarantee synchronization for generic initial conditions. Perturbations for which $\mathrm{Tr}(\delta\rho A) \neq 0$ generically

change the amplitude of the limit cycle due to the finite-dimensionality of the Hilbert space and linearity. This is an unavoidable implication of our results and constitutes an important fundamental difference between the linear time evolution of a quantum observable $O$ and the time evolution of a classical observable.

Secondly, stability to noise for classical synchronization follows directly from exponential stability - noise may be understood as a random series of perturbations. Quantum mechanically, the influence of fully generic noise is fundamentally different as it induces decoherence and relaxation to stationarity. We have shown in Th. 3 that quantum synchronization is stable to arbitrary noise/dissipation at least to the second-order in perturbation strength. Moreover, as we have shown, in the quantum case symmetry-selective (not arbitrary) noise is *fundamentally necessary* to induce synchronization.

Thirdly, the stability of the frequency of classical synchronization is *neutral*, which means that a perturbation can change the frequency, but in a way that neither grows nor decays in time [171]. In our case Th. 3 guarantees precisely this, and Cor. 4 shows that when a perturbation explicitly breaks the synchronization, the frequency is, counter-intuitively, one order more stable than to a perturbation that does not break synchronization. This elucidates quantum synchronization as a cooperative dynamical stabilization phenomenon analogous to classical synchronization.

# 5 Proving the Absence of Synchronization and Persistent Oscillations

We have provided a general theory of quantum synchronization based on necessary and sufficient algebraic conditions that may naturally be applied to quantum many-body systems. The following question remains: how can one easily show an absence of synchronization in a quantum system? Apart from calculating the long-time dynamics or diagonalizing the Liouvillian, which may be analytically or even numerically intractable for an extensive system, we can use the theory we have developed to provide the following theorem.

**Theorem 4.** *If there is a full rank stationary state $\rho_\infty$ (i.e. invertible) and the commutant[3] is proportional to the identity, $\{H, L_\mu, L_\mu^\dagger\}' = c\mathbb{1}$, then there are no purely imaginary eigenvalues of $\hat{\mathcal{L}}$ and hence no stable synchronization. If this holds in the leading order, there are no almost purely imaginary eigenvalues and no metastable synchronization.*

As shown in Appx. I, this is a consequence of the strong dynamical symmetry of Thm. 2 not being unitary for $\omega \neq 0$. The absence of such a non-trivial commutant is implied by $\{H, L_\mu, L_\mu^\dagger\}$ forming a complete algebra of $\mathcal{B}(\mathcal{H})$. This can be shown straightforwardly in numerous cases. For instance, if we have a system of $N$ spin-1/2 and $L_k^+ = \gamma_k \sigma_k^+$, $L_k^- = \gamma_k \sigma_k^-$, and arbitrary $H$, the map is unital ($\mathbb{1}$ is a full rank stationary state) and e.g. $[L_k^+, (L_k^+)^\dagger] = \sigma_k^z$ form a complete algebra of Pauli matrices on $\mathcal{B}(\mathcal{H})$. Further, the results of Evans and Frigerio [172–175] tell us that a trivial commutant is equivalent to the stationary state being unique.

As a broad set of examples to which this idea can be applied, under the mild assumption of having a full rank NESS, any 1D spin-1/2 system with nearest-neighbour interaction undergoing Markovian dissipation modelled by on-site non-Hermitian Lindblad operators cannot have purely imaginary eigenvalues. This follows from our result and the construction of the complete algebra given in Sec. 2.1 of [176]. Note that the spin-1/2 example we give later does not have a full rank NESS.

---

[3]The commutant of a set of linear operators on a Hilbert space $\mathcal{A} \subset \mathcal{B}(\mathcal{H})$, denoted $\mathcal{A}'$, is the set of $O \in \mathcal{B}(\mathcal{H})$ which commute with all $A \in \mathcal{A}$. Note that all multiplies of the identity trivially belong to the commutant of any set.

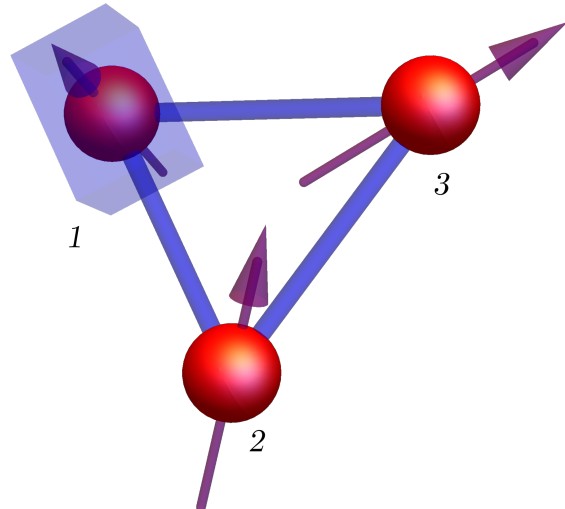

Figure 3: Anti-synchronizing two spin-1/2 (sites 2, 3) via a non-Markovian bath. The bath is the site 1 spin and the $L = \gamma \sigma_1^-$ loss term (the blue box).

## 6 Examples

To demonstrate the application of our theory and to showcase the results of Sections 3 and 4 we now present two examples. The first is a straightforward demonstration of how to anti-synchronize two qubits using a third ancilla qubit. While being an interesting result in its own right, since it has previously been implied that two qubits cannot synchronize in the sense of [38], this example should provide a pedagogical explanation of how our theory works. We then move on to discuss the Hubbard model with spin-agnostic heating. This example has been used several times to study long-time non-stationarity in open quantum systems, and here we use it as a springboard to construct a family of more generalized models that will exhibit quantum synchronization. These generalized models are of interest for their relation to experiments involving particles of spin $S > 1/2$. Note that there are further applications of our theory to analyze additional examples from existing literature in the Appendices.

### 6.1 Perfectly Anti-Synchronizing Two Spin-1/2's

In [38] it was argued that the smallest possible system that can be synchronized is a spin-1, and this was then extended in [37] to two spin-1's. Here, using our theory, we complement this result by showing how it is, in fact, possible to *anti-synchronize* two spin-1/2's through what is effectively a non-Markovian bath. Let $\sigma_j^\alpha, \alpha = +, -, z$ be the standard Pauli matrices on the site $j$. Take any 3-site Hamiltonian which non-trivially couples the three sites, is reflection symmetric around the first site, $P_{2,3} H P_{2,3} = H$, and further conserves total magnetization $S^z = \frac{1}{2} \sum_j \sigma_j^z, [H, S^z] = 0$. The Hamiltonian can then be decomposed into blocks of conserved total magnetization. Furthermore, the block with eigenvalue $S^z = -1/2$, i.e. one spin-up and the other two spin-down, has one eigenstate, $|\phi\rangle$, with a node on site 1, i.e. $\sigma_1^- |\phi\rangle = 0$. To see this explicitly, consider the most general ansatz for an eigenstate with just a single excitation,

$$|\psi\rangle = \sum_j a_j |j\rangle, \tag{30}$$

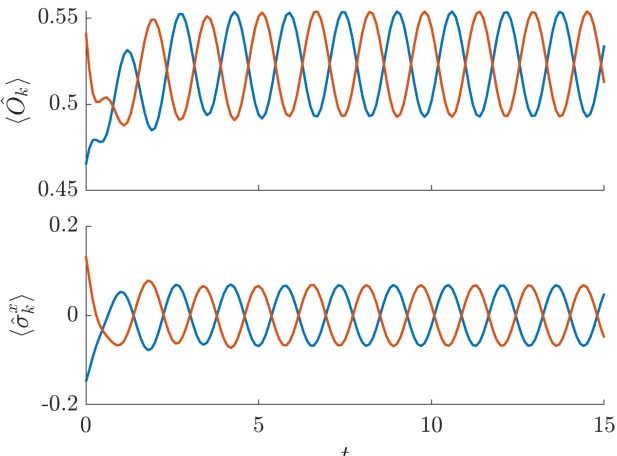

Figure 4: Evolution of the model described in Sec. 6.1 with parameters $\Delta = 1$, $B = 0.5$ and $\gamma = 2$. The system is initially described by a randomly chosen density matrix. In the top plot we compare a randomly generated Hermitian observable on sites 2 and 3 (blue and red curves respectively). While they oscillate out of phase they have an offset equilibrium value which disrupts the perfect anti-synchronization. In the bottom plot we now compare the $\sigma^x$ observable on each site and see perfect anti-synchronization since $\langle \sigma_1^z \rangle \to 0$ as $t \to \infty$.

where $j$ means there is a spin-up on-site $j$ and all other spins are down. Then to be an eigenstate with $H |\psi\rangle = E |\psi\rangle$ we see,

$$
\begin{aligned}
H(P_{2,3} |\psi\rangle) &= P_{2,3} H P_{2,3} P_{2,3} |\psi\rangle \\
&= P_{2,3} H |\psi\rangle \\
&= E P_{2,3} |\psi\rangle \\
&= E(P_{2,3} |\psi\rangle).
\end{aligned}
\tag{31}
$$

Thus

$$
\begin{aligned}
|\phi\rangle &= |\psi\rangle - P_{2,3} |\psi\rangle \\
&= (a_2 - a_3) |2\rangle + (a_3 - a_2) |3\rangle,
\end{aligned}
\tag{32}
$$

is an eigenstate of H with energy $E$ and it has a node on site 1. Provided the Hamiltonian non-trivially couples the three sites, this state will be the unique eigenstate in the $S^z = -1/2$ sector with a node on site 1. We may exploit this by considering site 1 with a pure loss Lindblad $L = \gamma \sigma_1^-$ as the bath and the sites 2 and 3 as the system. This is illustrated in Fig. 3.

In this case we have two stationary states,

$$
\rho_{1,\infty} = |0,0,0\rangle \langle 0,0,0|
\tag{33}
$$

$$
\rho_{2,\infty} = \frac{1}{2}(|0,0,1\rangle - |0,1,0\rangle)(\langle 0,0,1| - \langle 0,1,0|).
\tag{34}
$$

Note that these are pure states and form a decoherence-free subspace [177]. The $A$ operator satisfying Th. 1 is,

$$
A = (|0,0,1\rangle - |0,1,0\rangle) \langle 0,0,0| .
\tag{35}
$$

The frequency $\omega$ will depend on the specific choice of the Hamiltonian. Taking, for example, the XXZ spin chain,

$$
H = \sum_{j=1}^{3} \sigma_j^+ \sigma_{j+1}^- + \sigma_j^- \sigma_{j+1}^+ + \Delta \sigma_j^z \sigma_{j+1}^z + B \sigma_j^z,
\tag{36}
$$

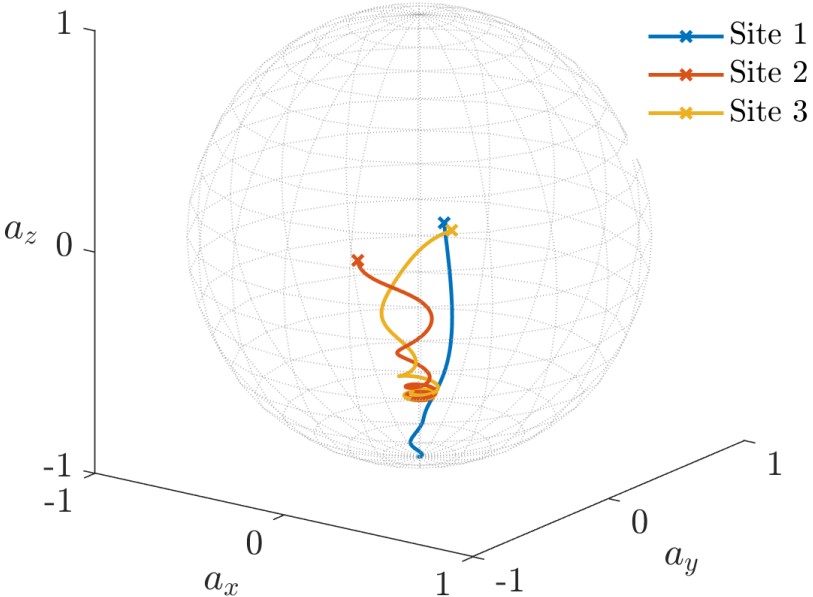

Figure 5: Bloch sphere representation of the evolution of the model described in Sec. 6.1 with parameters $\Delta = 1$, $B = 0.5$ and $\gamma = 2$ and a randomly chosen initial state. We define the reduced density matrix for each site by taking the partial trace over the other sites, $\rho_k = \text{Tr}_{A,B \neq k}(\rho)$. We then find and plot the corresponding Bloch sphere representation, $\vec{a}^{(k)}$ for the reduced states as $\rho_k = \frac{1}{2}(\mathbb{I} + \vec{a}^{(k)} \cdot \vec{\sigma})$ where $\vec{\sigma}$ are the usual Pauli matrices. The initial point of each trajectory is marked with a cross. We see the second and third sites reach a limit cycle which they orbit perfectly out of phase, while the first site rapidly decays to the $\vec{a} = (0, 0, 1)$ point on the Bloch sphere. This demonstrates the anti-synchronization between only sites 2 and 3. Note that since the reduced states are note pure the trajectories live within the sphere.

with implied periodic boundary conditions, we have $\omega = -1 + 2B - 4\Delta$. Note that there are persistent oscillations even in the absence of an external field $B = 0$. In that case the interaction term $\sigma_j^z \sigma_{j+1}^z$ (corresponding in the Wigner-Jordan picture to a quartic term) picks out a natural synchronization frequency. Since $A$ is antisymmetric, $P_{2,3} A P_{2,3} = -A$, the oscillating coherences will also be antisymmetric. The symmetric stationary state $\rho_{1,\infty}$ spoils anti-synchronization between site 2 and 3 by offsetting the equilibrium value. However, an observable that is zero in this state $\text{tr}(O_k \rho_{1,\infty}) = 0$, $k = 2, 3$, will be robustly and stably anti-synchronized $\lim_{t \to \infty} \langle O_2(t) \rangle = -\lim_{t \to \infty} \langle O_3(t) \rangle$ with frequency $\omega$. A possible choice is the transverse spin $O_k = \sigma_k^x$, $k = 2, 3$. This is demonstrated in Figure 4.

Using this example, we can further make a link to previous studies of quantum synchronization, which focused on limit cycles in phase space. In Figure 5 we show the limit cycles of the second and third spins in the phase space defined by the usual Bloch sphere representation of a qubit. We see that the two limit cycles are perfectly out of phase, as expected for anti-synchronization. We can also see clearly that the first site is not synchronized to either the first or second site since its phase space trajectory quickly decays to a fixed point rather than the common limit cycle of sites 2 and 3.

## 6.2 Lattice Models with Translationally Invariant non-Abelian Symmetries: Stable and Metastable Synchronization

We will now proceed to explore examples of many-body models that exhibit robust and stable/meta-stable synchronization between every site due to their symmetry structure. We first discuss the Fermi-Hubbard model as a base case and then explore its generalizations to multi-band and higher spin versions before commenting on how our theory can be applied to existing experimental set-ups. We will also explain how systems can be engineered using the theory in this work to create long-lived synchronization.

### 6.2.1 Fermi-Hubbard Model with Spin Agnostic Heating

The standard Fermi-Hubbard model has been previously shown to exhibit synchronization in the presence of spin agnostic heating [36, 54, 57]. We can now understand the appearance of synchronization in this model as a consequence of Cor. 3 as follows.

Consider a generalized $N$-site, spin-1/2 Fermi-Hubbard model on any bipartite lattice,

$$
\begin{aligned}
H = -\sum_{\langle i,j \rangle} \sum_{s \in \{\uparrow, \downarrow\}} (c_{i,s}^\dagger c_{j,s} + \text{h.c}) + \sum_j U_j n_{j,\uparrow} n_{j,\downarrow} \\
+ \sum_j \epsilon_j n_j + \frac{B_j}{2} (n_{j,\uparrow} - n_{j,\downarrow}),
\end{aligned}
\tag{37}
$$

where $c_{j,s}$ annihilates a fermion of spin $s \in \{\uparrow, \downarrow\}$ on site $j$ and $\langle i,j \rangle$ denotes nearest neighbor sites. The particle number operators are $n_{j,s} = c_{j,s}^\dagger c_{j,s}$, $n_j = \sum_s n_{j,s}$. This model includes on-site interactions $U_j$, a potential $\epsilon_j$ and an inhomogenous magnetic field $B_{j,s}$ in the $z$-direction. We further include dominant standard 2-body loss, gain and dephasing processes, naturally realized in optical lattice setups [178–183],

$$
L_j^- = \gamma_j^- c_{j,\uparrow} c_{j,\downarrow}
\tag{38}
$$

$$
L_j^+ = \gamma_j^+ c_{j,\uparrow}^\dagger c_{j,\downarrow}^\dagger
\tag{39}
$$

$$
L_j^z = \gamma_j^z n_j,
\tag{40}
$$

with $j = 1, \ldots N$.

From previous studies relating to time crystals, [54, 57, 59] it is now well appreciated that in a homogenous magnetic field $B_j = B$ the total spin raising operator $S^+ = \sum_j c_{j,\uparrow}^\dagger c_{j,\downarrow}$ is a strong dynamical symmetry with,

$$
[H, S^+] = B S^+, \quad [L_j^\alpha, S^+] = 0.
\tag{41}
$$

For $\gamma_j^- = \gamma_j^+$ we have

$$
\sum_{\alpha, j} [L_j^\alpha, (L_j^\alpha)^\dagger] = 0,
\tag{42}
$$

so we find the map is unital. Provided that $S^+$ and its conjugate $S^-$ are the only operator satisfying the conditions of Th. 2, as is generically the case unless $\gamma_j^- = \gamma_j^+ = \gamma_j^z = 0$, we may apply Cor. 3 and conclude that every site is stably, robustly synchronized with every other site because $S^+$ and $S^-$ have complete permutation invariance. We emphasise that the on-site potentials, $\epsilon_j$, need not be the same in this case. This explicitly demonstrates how our theory can be applied non-perturbatively to non-homogeneous systems.

Metastability can be achieved by, for example, allowing detuning between the on-site magnetic field and considering perturbations from the average value of $B_j$ $\bar{B} := 1/N \sum_j B_j$, i.e. $s_j = B_j - \bar{B}$. As per Cor. 4 the synchronization will be stable to second order, i.e. $\sim \max(s_j^2)$.

This model hints at a general framework to construct more elaborate examples of systems that exhibit quantum synchronization based on their symmetries. We give details on this general framework in Appx. M, which can in principle be applied to a broad class of models, including fermionic tensor models relevant for high energy physics and gauge/gravity dualities, [184], and $U(2N)$ fermionic oscillators [101]. We will now discuss the case of multi-band, and $SU(N)$ Hubbard models, which have previously been explored in cold atoms [185–188].

### 6.3 Generalised Fermi-Hubbard Model with $SU(N)$ Symmetry and Experimental Applications

For concreteness, we will first consider the model studied in [188] describing fermionic alkaline-earth atoms in an optical lattice. These atoms are often studied as they have a meta-stable $^3P_0$ excited state, which is coupled to the $^1S_0$ ground state through an ultra-narrow doubly-forbidden transition [189]. We will refer to these levels as $g$ (ground) and $e$ (excited). We will further label the nuclear Zeeman levels as $m = -I, \ldots, I$ on-site $i$, where $N = 2I + 1$ is the total number of Zeeman levels of the atoms. For example, $^{87}$Sr has $N = 10$. It is further known that in these atoms, the nuclear spin is almost completely decoupled from the electronic angular momentum in the two states $\{g, e\}$ [189]. Thus to a good level of approximation, one can describe a system of these atoms in an optical trap using a two-orbital single-band Hubbard Hamiltonian [188],

$$
\begin{aligned}
H = &-\sum_{\langle i,j \rangle} \sum_{s,m} J_s(c^\dagger_{i,s,m} c_{j,s,m} + c^\dagger_{j,s,m} c_{i,s,m}) \\
&+ \sum_{j,s} U_{ss} n_{j,s}(n_{j,s} - 1) + V \sum_j n_{j,g} n_{j,e} \\
&+ V_{ex} \sum_{j,m,m'} c^\dagger_{j,g,m} c^\dagger_{j,e,m'} c_{j,g,m'} c_{j,e,m}.
\end{aligned}
\tag{43}
$$

Here the $c_{i,s,m}$ operator annihilates a state with nuclear spin $m$ and electronic orbital state $s \in \{g, e\}$ on site $i$. Further $n_{j,s} = \sum_m c^\dagger_{j,s,m} c_{j,s,m}$ counts the number of atoms with electronic orbital state $s$ on site $j$. This model assumes that the scattering and trapping potential are independent of nuclear spin, which gives rise to a large $SU(N)$ symmetry.

Defining the nuclear spin permutation operators as

$$
S^m_n = \sum_{j,s} c^\dagger_{j,s,n} c_{j,s,m},
\tag{44}
$$

which obey the $SU(N)$ algebra,

$$
[S^m_n, S^p_q] = \delta_{mq} S^p_n - \delta_{np} S^m_q,
\tag{45}
$$

we find that the Hamiltonian in equation (43) has full $SU(N)$ symmetry,

$$
[H, S^m_n] = 0 \;\forall n, m.
\tag{46}
$$

We can also introduce the electron orbital operators as,

$$
T^\alpha = \sum_{j,s,s',m} c^\dagger_{j,s,m} \sigma^\alpha_{ss'} c_{j,s',m},
\tag{47}
$$

where $\alpha = x, y, z$ and $\sigma^\alpha$ are the Pauli matrices in the $g, e$ basis. These operators obey the usual $SU(2)$ algebra and are independent of the nuclear spin operators, $[T^\alpha, S^m_n] = 0$. In the specific case $J_e = J_g$, $U_{ee} = U_{gg} = V$, $V_{ex} = 0$ these are also a full symmetry of the system, $[H, T^\alpha] = 0$.

If we now introduce dephasing of the nuclear spin levels via the on-site Lindblad operators

$$L_j^{(m)} = \gamma_j^{(m)} S_m^m, \tag{48}$$

we break the $SU(N)$ nuclear-spin symmetry, while the electronic orbital symmetry is promoted to a strong symmetry [159] since the operators $T^\alpha$ all commute with the Lindblad operators. This may be accomplished by scattering with incoherent light that does not distinguish between various energy levels of the internal degree of freedom $s$ [178, 190]. In particular, this represents the fact that the light has transmitted information to the environment about the location of an atom, but not its internal degree of freedom that remains coherent. The rates $\gamma_j^{(m)}$ can be made larger by introducing more light scattering.

In the presence of a field that only couples to the electronic degrees of freedom,

$$H_{\text{field}} = \omega T^z, \tag{49}$$

the electronic orbital $SU(2)$ symmetry is broken to a dynamical symmetry with frequency $\omega$. Since the $T^\pm$ operators are translationally independent, all the conditions of Cor. 3 are satisfied to guarantee robust, stable synchronization between all pairs of sites.

In realistic set-ups, this strict symmetry structure is unlikely to be perfectly maintained. In particular certain cold atom species may have a finite exchange term $V_{ex}$ between the spin levels, e.g. [191, 192]. Thus we would expect metastable synchronization to be present when the conditions $J_e = J_g$, $U_{ee} = U_{gg} = V$, $V_{ex} = 0$ do not hold perfectly or there are some inhomogeneities in the additional field. More generally, we could consider cases of interacting systems where the nuclear and electronic degrees of freedom are coupled through scattering processes, such as in the experiment conducted by [144]. Generically such systems will lack the symmetries required for robust, stable quantum synchronization, and thus the timescale over which synchronization can be maintained will be a measure of how much the symmetries are broken through these imperfections.

One possible approach for achieving metastable synchronization in more complex systems is to proceed as follows. Ignoring interactions, we can diagonalize the single site Hamiltonians to obtain eigenstates $|n\rangle_j$ and energy levels $E_{n,j}$ for each site, $j$. These give trivial onsite dynamical symmetries $A_j^{n,m} = |n\rangle \langle m|_j$ for $E_{n,j} \neq E_{m,j}$. We then introduce dephasing operators on each site which break all but a few of these dynamical symmetries. When we reintroduce interactions between neighbouring sites, we search for translationally invariant linear combinations of the remaining on-site strong dynamical symmetries, which are as close as possible to dynamical symmetries of the whole model. These linear combinations can, in principle, be further optimized by adjusting the experimental parameters of the system. These considerations emphasize the importance of our theory when engineering long-lived synchronization in more complex systems. Our theory tells us that to produce quantum systems that exhibit long-lived synchronization, the symmetries must be carefully controlled.

Another aspect of our theory that can be applied to these more complicated systems, even if they do not admit the symmetries required for synchronization, is to give the scalings of decay rates and frequencies of the meta-stable oscillations. As an example, we consider the experimental set-up investigated by [144, 193], where fermionic atoms with nuclear spin-9/2 were confined to a deep optical lattice. In their experiment, they initialized the system so that every site contained two atoms, one with $m = 9/2$ and the second with $m = 1/2$, and observed oscillations across the whole system between this initial state and the state in which the two atoms had spins $m = 7/2$ and $m = 3/2$. These oscillations were most pronounced in the limit where the optical lattice was very deep, corresponding to minimal hopping. It is known that deep optical lattices can often cause dephasing processes to occur, so it is likely that in the limit of no hopping, the system also experiences strong dephasing processes. Thus we apply

the results of Sec. 4.2 to predict that as the trap becomes shallower, the frequency should vary as,

$$\omega = \omega_0 + \frac{\lambda}{\gamma} + o(1/\gamma), \tag{50}$$

for some constant $\lambda$, where $\gamma = U/J$ is the ratio between the interaction strength and the hopping. This ratio is known to be related to the lattice depth in a 1D sinusoidal lattice in the deep lattice limit as [194],

$$\gamma \sim \exp(\sqrt{V_0}), \tag{51}$$

for $V_0$ the lattice depth, measured in units of $E_r$. The 1D approximation is valid because the other two dimensions of the optical lattice are kept at very deep values $V_\perp = 35E_r$ [144]. Thus we obtain

$$\omega(V_0) = \omega_0 + \lambda \exp(-\sqrt{V_0}). \tag{52}$$

Our simple results appear to be in better agreement with the experimental measurements than the numerical simulations in [144].

# 7 Conclusions

We have introduced a general theory of quantum synchronization in many-body systems without well-defined semi-classical limits. So far, although a subject of great interest, quantum synchronization has been studied on an ad-hoc basis without a systematic framework. The advantage of our theory is that it provides an algebraic framework based on dynamical symmetries from which to systematically study quantum synchronization in many-body systems. Symmetries are useful in quantum physics, especially many-body physics, as they allow for exact results and statements without resorting to challenging and often infeasible analytical or numerical computations. Our framework also allows for exact solutions of all such dynamics in terms of this dynamical symmetry algebra.

We introduced definitions of stable and metastable synchronization, which align with the notion of identical synchronization in classical dynamical systems. Stable synchronization lasts for an infinite amount of time, whereas metastable lasts only for very long times compared to the system's characteristic timescale and is the one usually studied in the literature. In addition, we defined robust synchronization to capture the robustness of the synchronized behaviour to the initial conditions. This is important to identify systems where the synchronization process is performed by some internal mechanism rather than fine-tuning of the initial state. We further divided the cases of metastable synchronization into those for which the observable dynamics take place on relevant time scales or not. We have demonstrated the synchronization is associated with quantum coherence.

We provided several examples, both new ones and from existing literature, and have shown how to use our theory to understand and extend them. Curiously, even though it was implied that the smallest system that can be synchronized is a spin-1 [38], we have used our theory to find an example of two spin-1/2's that anti-synchronize through interaction with a non-Markovian bath without contradicting the results of [38]. We then studied the Fermi-Hubbard model and its generalizations to explore how to generate models which can be expected to exhibit quantum synchronization. We also discussed how these results relate to experimental set-ups and demonstrated why robust quantum synchronization requires careful engineering and controlling experimental imperfections. Apart from higher symmetry fermionic quantum gases we discussed, similar considerations can also be directly applied to other complex cold atom systems with high degrees of symmetry and a large number of degrees of freedom, such as quantum spinor gases [195–197]. This demonstrates how our theory provides a guideline

for achieving synchronization that would be difficult to predict without using an algebraic perspective.

Our results provide several illuminating insights which help generate models that exhibit synchronization. The first is that the most straightforward way to synchronize quantum many-body systems is to use unital maps, e.g. dephasing. This is because we may then reduce the problem to eliminating dynamical symmetries that lack the required permutation symmetry structure for synchronization. Our results also indicate the importance of interactions. For instance, if a model has only quadratic 'interacting' terms corresponding to hopping or on-site fields, dynamical symmetries that are not translationally invariant are possible [36]. In particular free-fermion models admit a host of non-translationally invariant conservation laws $[H, Q_k] = 0$, and these ruin the translation invariance in the long-time limit [198]. Adding interactions generally leaves only translationally invariant $A$ operators.

We also note that there has been recent debate about the related phenomenon of limit cycles in driven-dissipative systems with finite local Hilbert space dimensions (in particular spin-1/2 systems). Mean-field methods find evidence of limit cycles (e.g. [199]), whereas including quantum correlations with some numerical methods seems to indicate an absence of limit cycle phases in these models (e.g. [200, 201]). However, other methods [202] accounting for quantum correlations do show limit cycles, making the issue controversial. We note that limit cycles correspond to persistent dynamics and purely imaginary eigenvalues of the corresponding quantum Liouvillians. Thus our general algebraic theory should apply to these systems and can be used to prove either presence or absence of limit cycles.

One direct generalization of our work should be to quantum Liouvillians with an explicit time dependence that would allow for the study of synchronization to an external periodic drive. In this case, one could also study discrete time translation symmetry breaking and discrete time crystals under dissipation [203, 204], which would correspond to purely imaginary eigenvalues in the Floquet Liouvillian or equivalently to eigenvalues of the corresponding propagator lying on the unit circle. A further direction that should be explored is the synchronisation of sites which have differing local Hilbert space dimension, such as synchronising a spin-1/2 with a spin-1. Our framework should be applied in the future for generating new models that have synchronization. Finally, extensions to more general types of synchronization such as amplitude envelope synchronization and those models with infinite-dimensional Hilbert spaces should also follow.

# Acknowledgements

We thank C. Bruder and G. L. Giorgi for useful discussions.

**Author contributions**  BB conceived the research and stated and proved the main theorems. CB stated and proved some of the theorems and wrote the majority of the manuscript. DJ suggested experimentally relevant examples. All authors contributed to discussions and writing of the manuscript.

**Funding information**  We acknowledge funding from EPSRC programme grant EP/P009565/1, EPSRC National Quantum Technology Hub in Networked Quantum Information Technology (EP/M013243/1), and the European Research Council under the European Union's Seventh Framework Programme (FP7/2007-2013)/ERC Grant Agreement no. 319286, Q-MAC. The work of DJ was partly supported by the Deutsche Forschungsgemeinschaft (DFG, German Research Foundation) via Research Unit FOR 2414 under project number 277974659 and via the Cluster of Excellence 'CUI: Advanced Imaging of Matter'—EXC

2056—under project number 390715994.

# Appendices

We now provide proofs of the results in Sections 3 and 4 along with additional analysis of previously studied examples of quantum synchronization and a discussion of how to construct more elaborate models which exhibit quantum synchronization using symmetry structures.

## A   Proof of Theorem 1

**Theorem.** *The following condition is necessary and sufficient for the existence of an eigenstate $\rho$ with purely imaginary eigenvalue $i\lambda$, $\hat{\mathcal{L}}\rho = i\lambda\rho$, $\lambda \in \mathbb{R}$.*

*We have $\rho = A\rho_\infty$, where $\rho_\infty$ is a NESS and $A$ is a unitary operator which obey,*

$$[L_\mu, A]\rho_\infty = 0, \tag{53}$$

$$\left(-i[H,A] - \sum_\mu [L_\mu^\dagger, A]L_\mu\right)\rho_\infty = i\lambda A\rho_\infty, \ \lambda \in \mathbb{R}. \tag{54}$$

*Proof.* Sufficiency can be checked directly by calculating $\hat{\mathcal{L}}[\rho] = \hat{\mathcal{L}}[A\rho_\infty]$. To prove the converse we first observe that $\rho$ is also an eigenstate of the corresponding quantum channel $\hat{\mathcal{T}}_t = e^{t\hat{\mathcal{L}}}$ with eigenvalue $e^{i\lambda t}$. Since this lies on the unit circle we may apply Theorem 5 of [156] to deduce that $\rho$ admits a polar decomposition of the form $\rho = AR$ where $A$ is unitary and $R$ is positive semi-definite with $\hat{\mathcal{T}}_t R = R$. In particular this implies $\hat{\mathcal{L}}[R] = 0$ so that $R$ is a steady-state of $\hat{\mathcal{L}}$ which we now call $\rho_\infty = R$. Note that this also implies $\rho_\infty$ is Hermitian and may be scaled to have unit trace.

Writing the channel in Kraus form as

$$\hat{\mathcal{T}}_t[x] = \sum_k M_k(t)x M_k^\dagger(t), \tag{55}$$

we can apply Theorem 5 of [156] again to find

$$M_k(t)A\rho_\infty = e^{i\lambda t}AM_k(t)\rho_\infty. \tag{56}$$

Now note that the adjoint channel is given by $\hat{\mathcal{T}}_t^\dagger[x] = \sum_k M_k(t)^\dagger x M_k(t)$, and so we can compute the adjoint Liouvillian as,

$$\begin{aligned}\hat{\mathcal{L}}^\dagger[x] &= \frac{d\hat{\mathcal{T}}_t^\dagger}{dt}\Big|_{t=0} \\ &= \sum_k \dot{M}_k^\dagger(0)x M_k(0) + M_k^\dagger(0)x\dot{M}_k(0).\end{aligned} \tag{57}$$

Using the of derivative equation (56) and the requirement that Kraus operators satisfy

$\sum_k M_k^\dagger M_k = \mathbb{I}$ we can calculate,

$$
\begin{aligned}
\hat{\mathcal{L}}^\dagger[A]\rho_\infty &= \sum_k \dot{M}_k^\dagger(0)AM_k(0)\rho_\infty \\
&\quad + M_k^\dagger(0)A\dot{M}_k(0)\rho_\infty \\
&= \sum_k \dot{M}_k^\dagger(0)M_k(0)A\rho_\infty \\
&\quad + M_k^\dagger(0)\dot{M}_k(0)A\rho_\infty \\
&\quad + i\lambda M_k^\dagger(0)AM_k(0)\rho_\infty \\
&= \sum_k \frac{d}{dt}\big[M_k^\dagger(t)M_k(t)\big]_{t=0}A\rho_\infty \\
&\quad + i\lambda \sum_k M_k^\dagger(0)M_k(0)A\rho_\infty \\
&= i\lambda A\rho_\infty.
\end{aligned}
\tag{58}
$$

A similar calculation using the conjugate equations, noting that $\rho_\infty$ is Hermitian, yields

$$
\rho_\infty \hat{\mathcal{L}}^\dagger[A^\dagger] = -i\lambda \rho_\infty A^\dagger.
\tag{59}
$$

We will now also introduce the *dissipation function* [147, 157], defined for any operator $x$ as

$$
\begin{aligned}
D[x] &= \hat{\mathcal{L}}^\dagger[x^\dagger x] - \hat{\mathcal{L}}^\dagger[x^\dagger]x - x^\dagger \hat{\mathcal{L}}^\dagger[x] \\
&= \sum_\mu [L_\mu, x]^\dagger [L_\mu, x].
\end{aligned}
\tag{60}
$$

Then by unitarity of $A$ and the above results for $\hat{\mathcal{L}}^\dagger[A]$, $\hat{\mathcal{L}}^\dagger[A^\dagger]$ we compute

$$
0 = \rho_\infty D[A]\rho_\infty = \sum_\mu \rho_\infty [L_\mu, A]^\dagger [L_\mu, A]\rho_\infty.
\tag{61}
$$

Since this sum is positive definite we must have $[L_\mu, A]\rho_\infty = 0 \ \forall \mu$. We now compute $\hat{\mathcal{L}}[A\rho_\infty] = i\lambda A\rho_\infty$ using $\hat{\mathcal{L}}[\rho_\infty] = 0$ and $[L_\mu, A]\rho_\infty = 0$ to obtain

$$
\left(-i[H,A] - \sum_\mu [L_\mu^\dagger, A]L_\mu\right)\rho_\infty = i\lambda A\rho_\infty,
\tag{62}
$$

as required. $\qquad\square$

# B Proof of Corollary 1

**Corollary.** *The Liouvillian super-operator, $\hat{\mathcal{L}}$, admits a purely imaginary eigenvalue, $i\lambda$ only if there exists some unitary operator, $A$, satisfying*

$$-i\rho_\infty A^\dagger[H,A]\rho_\infty = i\lambda\rho_\infty^2, \tag{63}$$

$$\rho_\infty A^\dagger[L_\mu^\dagger,A]L_\mu\rho_\infty = 0, \; \forall\mu, \tag{64}$$

$$\rho_\infty[L_\mu^\dagger,A^\dagger]L_\mu\rho_\infty = 0, \; \forall\mu. \tag{65}$$

*In which case the eigenstate is given by $\rho = A\rho_\infty$.*

*Proof.* We may use the unitarity of $A$ and $[L_\mu,A]\rho_\infty = 0 \; \forall\mu$ to easily obtain

$$\rho_\infty A^\dagger[L_\mu^\dagger,A]L_\mu\rho_\infty = 0, \; \forall\mu. \tag{66}$$

We then left multiply equation (62) by $\rho_\infty A^\dagger$ and use (66) to obtain

$$-i\rho_\infty A^\dagger[H,A]\rho_\infty = i\lambda\rho_\infty^2 \tag{67}$$

as stated. We get $\rho_\infty[L_\mu^\dagger,A^\dagger]L_\mu\rho_\infty = 0, \forall\mu$ by using (66) and the unitary of $A$, $[L_\mu^\dagger,A^\dagger A] = 0$
□

# C Proof of Theorem 2

**Theorem.** *When there exists a faithful (i.e. full-rank/ invertible) stationary state, $\tilde{\rho}_\infty$, $\rho$ is an eigenstate with purely imaginary eigenvalue if and only if $\rho$ can be expressed as,*

$$\rho = \rho_{nm} = A^n\rho_\infty(A^\dagger)^m, \tag{68}$$

*where $A$ is a (not necessarily unitary) strong dynamical symmetry obeying*

$$\begin{aligned}[H,A] &= \omega A \\ [L_\mu,A] = [L_\mu^\dagger,A] &= 0, \; \forall\mu\end{aligned} \tag{69}$$

*and $\rho_\infty$ is some NESS, not necessarily $\tilde{\rho}_\infty$. Moreover the eigenvalue takes the form*

$$\lambda = -i\omega(n-m). \tag{70}$$

*Furthemore, the corresponding left eigenstates, with $\hat{\mathcal{L}}^\dagger\sigma_{mn} = i\omega(n-m)\sigma_{mn}$, are given by $\sigma_{mn} = (A')^m\sigma_0\left((A')^\dagger\right)^n$ where $A'$ is also a strong dynamical symmetry and $\sigma_0 = \mathbb{1}$.*

*Proof.* We will make use of [156], [139] and [157]. The asymptotic subspace of the Liouvillian $As(\mathcal{H})$ is defined as a subspace of the space of linear operators $\mathcal{B}(\mathcal{H})$ such that all initial states $\rho(0)$ in the long-time limit end up in $\rho(t\to\infty)\in As(\mathcal{H})$. The projector $P$ ($P^2 = P = P^\dagger$) to the corresponding non-decaying part of the Hilbert space $\mathcal{H}$ is uniquely defined [139,157] as, for all $\rho(t\to\infty)\in As(\mathcal{H})$,

$$\begin{aligned}\rho(t\to\infty) &= P\rho(t\to\infty)P \\ \text{tr}(P) &= \max_{\rho(t\to\infty)}\{\text{rank}\{\rho(t\to\infty)\}\}.\end{aligned} \tag{71}$$

It follows therefore if there is a full rank $\tilde{\rho}_\infty$ that $P = \mathbb{1}$.

From the proof of Proposition 2 of [139] and Theorem 4 (Eqs. (2.39-2.40)) of [157] we know that for a left eigenmode with purely imaginary eigenvalue $A^\dagger \hat{\mathcal{L}} = -i\omega A^\dagger$, we have,

$$[PHP, PAP] = \omega PAP,$$
$$[PL_\mu P, PAP] = [PL_\mu^\dagger P, PAP] = 0. \tag{72}$$

Which reduces to

$$[H, A] = \omega A,$$
$$[L_\mu, A] = [L_\mu^\dagger, A] = 0. \tag{73}$$

Since $P = \mathbb{I}$. The left eigenmode $A^\dagger$ is also the eigenmode of the dual map $\hat{\mathcal{T}}_t^\dagger = \exp(\hat{\mathcal{L}}^\dagger t)$. Since $A^\dagger$ corresponds to a peripheral eigenvalue of $\hat{\mathcal{T}}_t^\dagger$, by Lemma 3 of [156], the corresponding right eigenmode with eigenvalue $i\omega$ is $\rho' = A\rho_\infty$. By the same Lemma, to every oscillating coherence $A\rho_\infty$ there corresponds a left eigenvector of the form $A\rho_\infty \tilde{\rho}_\infty^{-1} = A'$, which must also satisfy the conditions (73). Now we find that we can write $\rho' = A'\tilde{\rho}_\infty$ where $A'$ satisifes the conditions of (73). We now have the same criteria as those given in [54] and the statement of the theorem follows. The converse was shown in [54]. It is a straightforward calculation to show that $\sigma_{nm} = A^n (A^\dagger)^m$ is a left eigenmode with the desired eigenvalue. $\qquad\square$

## D  Proof of Corollary 2

**Corollary.** *The following conditions are sufficient for robust and stable synchronization between subsystems $j$ and $k$ with respect to the local operator $O$:*

- *The operator $P_{j,k}$ exchanging $j$ and $k$ is a weak symmetry [159] of the quantum Liouvillian $\hat{\mathcal{L}}$ (i.e. $[\hat{\mathcal{L}}, \hat{\mathcal{P}}_{j,k}] = 0$)*

- *There exists at least one $A$ fulfilling the conditions of Th. 1*

- *For at least one $A$ fulfilling the conditions of Th. 1 and the corresponding $\rho_\infty$ we have $\mathrm{tr}[O_j A\rho_\infty] \neq 0$*

- *All such $A$ fulfilling the conditions of Th. 1 also satisfy $[P_{j,k}, A] = 0$*

*Proof.*  In the long-time limit we have,

$$\lim_{t\to\infty} \langle O_j(t) \rangle = \mathrm{tr}\left[ O_j \sum_n c_n A_n \rho_{\infty,n} e^{i\lambda_n t} \right], \tag{74}$$

where $c_n = \langle\langle \sigma_n | \rho(0) \rangle\rangle$ for an initial state $\rho(0)$ which we assume to be arbitrary and $A_n$ are all operators satisfying the conditions of Th. 1 and $\hat{\mathcal{L}}\rho_{\infty,n} = 0$ forming a complete basis for the eigenspaces with purely imaginary eigenvalues. Clearly,

$$\lim_{t\to\infty} \langle O_j(t) \rangle = \mathrm{tr}\left[ O_j P_{j,k}^2 \sum_n c_n A_n \rho_{\infty,n} e^{i\lambda_n t} \right], \tag{75}$$

and we just need to commute the $P_{j,k}$ to the left to use $O_k = P_{j,k}O_j P_{j,k}$ which results in $\lim_{t\to\infty} \langle O_j(t) \rangle = \lim_{t\to\infty} \langle O_k(t) \rangle$ for any initial state. By assumption, $[A_n, P_{j,k}] = 0, \forall n$. We just need to show that $[\rho_{\infty,n}, P_{j,k}] = 0, \forall n$. To do so we use the fact that $P_{j,k}$ is a weak symmetry of the Liouvillian $[\hat{P}_{j,k}, \hat{\mathcal{L}}] = 0$. The two eigenvalues of $P_{j,k}$ are $+1, -1$ and it has orthogonal eigenspaces. This implies that the Liouvillian is block reduced to the eigenspaces $\mathcal{B}_1 = \{+1, +1\} \oplus \{-1, -1\}$ and $\mathcal{B}_2 = \{+1, -1\} \oplus \{-1, +1\}$ corresponding to the two $+1$ and $-1$

eigenvalues of $\hat{\mathcal{P}}_{j,k}$, respectively [159]. The only possible eigenmodes with eigenvalue 0 as per Theorem 18 of Baumgartner and Narnhofer [138] are either operators that are diagonal in some basis or *stationary phase relations*. The diagonal subspace $\mathcal{B}_1$ contains all matrices that are diagonal. To see this consider $\rho_2 \in \mathcal{B}_2$. It can be written as $\rho_2 = P_{+1}\rho_a P_{-1} + P_{-1}\rho_a P_1$ where

$$P_\pm = \frac{1}{2}\left(\mathbb{I} \pm P_{i,j}\right) \tag{76}$$

are the corresponding projectors which commute, $[P_{+1}, P_{-1}] = 0$, and are mutually diagonalizable. We have either $P_{+1}|\psi_+\rangle = |\psi_+\rangle$, $P_{-1}|\psi_+\rangle = 0$ or the reverse. Therefore $\langle\psi_a|\rho_2|\psi_a\rangle = 0$. For $\rho_{\infty,n} \in \mathcal{B}_1$ we immediately have $[\rho_{\infty,n}, P_{j,k}] = 0$. All the eigenmodes with eigenvalue 0 which belong to $\mathcal{B}_2$ are *stationary phase relations* because they only contain off-diagonal elements. It remains to show that all *stationary phase relations*, $\rho_{\infty,n} \in \mathcal{B}_2$, either satisfy $[\rho_{\infty,n}, P_{j,k}] = 0$ or vanish. This directly follows from Proposition 16 of Baumgartner and Narnhofer [138] that states the existence of a stationary phase relation implies the existence of a unitary $U$ such that $[H, U] = [L_\mu, U] = 0, \forall \mu$. However, such a $U$ must intertwine between the subspaces $+1$ and $-1$ of $P_{j,k}$, with projectors $P_1$ and $P_{-1}$, respectively, i.e. $UP_{+1} = P_{-1}U$ and therefore $[U, P_{j,k}] \neq 0$. However, $U$ satisfies all the conditions for an $A$ operator from Th. 1, and by assumption does not exist. Thus there are no non-zero $\rho_{\infty,n} \in \mathcal{B}_2$. □

# E  Proof of Corollary 3

**Corollary.** *The following conditions are sufficient for robust and stable synchronization between subsystems j and k with respect to the local operator O:*

- *The Liouvillian, $\hat{\mathcal{L}}$ is unital ($\hat{\mathcal{L}}(\mathbb{1}) = 0$)*

- *There exists at least one A fulfilling the conditions of Th. 1*

- *For at least on A fulfilling the conditions of Th. 1 and the corresponding $\rho_\infty$ we have $\mathrm{tr}[O_j A\rho_\infty] \neq 0$*

- *All such A fulfilling the conditions of Th. 1 also satisfy $[P_{j,k}, A] = 0$*

*Furthermore, if all A are translationally invariant, $[A, P_{j,j+1}] = 0, \forall j$, then every subsystem is robustly and stably synchronized with every other subsystem.*

*Proof.* We return to Eq. (75) and wish to show that $[\rho_{\infty,n}, P_{j,k}] = 0$ but without the assumption of weak symmetry. If $\hat{\mathcal{L}}$ is unital, then all $A_n$ satisfy conditions of Th.2 and therefore it straightforwardly follows that $A_n A_n^\dagger$ is a strong symmetry [159,205] of the Liouvillian and that the projectors to the eigenspaces of $A_n A_n^\dagger$, $P_{n,a}$, are stationary states $\hat{\mathcal{L}}P_{n,a} = 0$. By assumption $[P_{j,j+1}, A_n A_n^\dagger] = [P_{n,a}, P_{j,j+1}] = 0$. By Theorem 3 of [138] $P_{n,a}$ are projectors to enclosures. The existence of more minimal enclosures that do not satisfy the symmetry requirement would imply that there are more operators $A$ (as projectors to enclosures satisfy the conditions of Th. 1 trivially), and this cannot happen by assumptions of the corollary. Therefore, $P_{n,a}$ are projectors to minimal enclosures. By Theorem 18 of [138] all the minimal diagonal blocks $P_{n,a}\rho P_{n,a}$ contain a unique stationary state, which must be up to a constant $P_{n,a}$. The lack of stationary phase relations and oscillating coherences that do not commute with $P_{j,k}$ follows from the fact that by Theorem 18 of [138] the unique stationary state in each off-diagonal block is of the form of $UP_{n,a}$. This intertwiner, like in the proof of Cor. 2, satisfies the conditions for an $A$ and therefore must commute with $P_{j,k}$ by assumption. □

## F  Proof of Theorem 3

**Theorem.** *Let $\hat{\mathcal{L}}'(s)$ generate a CPTP map and depend analytically on s with Taylor series*

$$\hat{\mathcal{L}}'(s) = \hat{\mathcal{L}} + s\hat{\mathcal{L}}_1 + s^2\hat{\mathcal{L}}_2 + \mathcal{O}(s^2). \tag{77}$$

*Suppose there exists some $\omega \in \mathbb{R} \setminus \{0\}$ and $\rho_0$ such that $\hat{\mathcal{L}}\rho_0 = \mathrm{i}\omega\rho_0$. Then for s sufficiently small $\lambda(s)$ is an eigenvalue of $\hat{\mathcal{L}}'(s)$ given by*

$$\lambda(s) = \mathrm{i}\omega + \lambda_1 s + \lambda_2 s^{1+\frac{1}{p}} + o\left(\lambda_2 s^{1+\frac{1}{p}}\right), \tag{78}$$

*where $\lambda_1$ is purely imaginary, p is some positive integer and $\lambda_2$ has non-positive real part.*

*Proof.* We first note that if $\lambda(0)$ is a non-degenerate eigenvalue, then it is known that $\lambda(s)$ is analytic; thus, the above result is trivial. In fact, in this case, we also find that the eigenstate $\rho(s)$ also depends analytically on $s$.

For the case where $\mathrm{i}\omega$ is an $m$-fold degenerate eigenvalue of $\hat{\mathcal{L}}_0$ we define the 'ω-rotating stable manifold' (ω-RSM) as the eigenspace spanned by the $m$ eigenmode corresponding to $\mathrm{i}\omega$. We next prove the following lemma

**Lemma 1.** *The purely imaginary eigenvalues of $\hat{\mathcal{L}}$ have one-dimensional Jordan blocks.*

*Proof.* Let the CPTP map generated by $\hat{\mathcal{L}}$ be $\hat{\mathcal{T}}_t = e^{t\hat{\mathcal{L}}}$. Note that the eigenspace of $\hat{\mathcal{L}}$ with eigenvalue $\mathrm{i}\omega$ is also an eigenspace of $\hat{\mathcal{T}}_t$ with eigenvalue $e^{\mathrm{i}t\omega}$ which has unit modulus. Thus by proposition 6.1 of [206] the Jordan blocks corresponding to $e^{\mathrm{i}t\omega}$ in $\hat{\mathcal{T}}_t$ are all one-dimensional. It follows that the Jordan blocks of $\hat{\mathcal{L}}$ corresponding to $\mathrm{i}\omega$ are also all one-dimensional. $\square$

Since $\mathrm{i}\omega$ is a semi-simple eigenvalue and we can directly apply Theorem 2.3 of [207] to write

$$\lambda(s) = \mathrm{i}\omega + \lambda_1 s + \lambda_2 s^{1+\frac{1}{p}} + o\left(s^{1+\frac{1}{p}}\right) \tag{79}$$

for some integer $p \geq 1$. Finally we observe that since $\hat{\mathcal{L}}(s)$ always generates a CPTP map, we must have $\Re(\lambda(s)) \leq 0$ for all $s \in \mathbb{R}$ and thus immediately we can deduce that $\lambda_1$ is purely imaginary and $\lambda_2$ has non-positive real part. $\square$

## G  Proof of Corollary 4

**Corollary.** *For perturbations $\hat{\mathcal{L}}_1$ which explicitly break the exchange symmetry, and thus synchronization, between sites j and k, i.e. $\hat{\mathcal{P}}_{j,k}\hat{\mathcal{L}}_1\hat{\mathcal{P}}_{j,k} = -\hat{\mathcal{L}}_1$, the frequency of synchronization $\omega$ is stable to next-to-leading order in s, i.e. $\lambda_1 = 0$.*

*Proof.* We begin by noting that the projector to the $\lambda(s)$ group is differentiable to leading order (83). We may write the eigenmode equation $\hat{\mathcal{L}}'(s)\rho(s) = \lambda(s)\rho(s)$ in the first order as,

$$\hat{\mathcal{L}}\hat{\mathcal{P}}_1 + \hat{\mathcal{L}}_1\hat{\mathcal{P}}_0 = \lambda_0\hat{\mathcal{P}}_1 + \lambda_1\hat{\mathcal{P}}_0, \tag{80}$$

which we multiply by the corresponding left eigenvector of $\hat{\mathcal{L}}$ $\langle\langle\sigma(0)|$ and obtain,

$$\lambda_1 = \langle\langle\sigma(0)|\hat{\mathcal{L}}_1\hat{\mathcal{P}}_0, \tag{81}$$

which is clearly zero as $\hat{\mathcal{P}}_0 \in \mathcal{B}(\mathcal{B}_1)$, $\hat{\mathcal{L}}_1 \in \mathcal{B}(\mathcal{B}_{-1})$ and $\langle\langle\sigma(0)|\in \mathcal{B}_1$, where the indices of $\mathcal{B}_p$ denote the $p = \pm 1$ eigenspaces of the exchange superoperator $\hat{\mathcal{P}}_{j,k}$. $\square$

## H Proof of Corollary 5

**Corollary.** *Suppose our Liouvillian $\hat{\mathcal{L}}'(s)$ has Hamiltonian $H(s)$ and jump operators $L_\mu(s)$. If the perturbation is such that*

$$H(s) = H^{(0)} + sH^{(1)} + \mathcal{O}(s^2)$$
$$L_\mu(s) = L_\mu^{(0)} + \mathcal{O}(s^2) \tag{82}$$

*then we find that $p = 1$ in the result of Theorem 3.*

*Proof.* We follow the reduction method of [207] and [166]. From [207] we recall that the eigenvalues perturbed away from $\omega$ (called the $\omega$-group) are not in general analytic. They are instead branches of analytic functions and the corresponding eigenstates may contain poles. However the projection onto the span of the $\omega$-group eigenstates is analytic and thus the restriction of $\hat{\mathcal{L}}(s)$ to this subspace is also analytic. Let us write this projection operator as $\hat{\mathcal{P}}(s)$ and thus the restricted Liouvillian is given by $[\hat{\mathcal{L}}(s)]_{\hat{\mathcal{P}}(s)} = \hat{\mathcal{P}}(s)\hat{\mathcal{L}}(s)\hat{\mathcal{P}}(s)$. We can use the result from Section II.2 of [207] to write

$$\hat{\mathcal{P}}(s) = \hat{\mathcal{P}} + s\hat{\mathcal{P}}_1 + \mathcal{O}(s^2), \tag{83}$$

where

$$\hat{\mathcal{P}}_1 = -\hat{\mathcal{S}}\hat{\mathcal{L}}_1\hat{\mathcal{P}} - \hat{\mathcal{P}}\hat{\mathcal{L}}_1\hat{\mathcal{S}}. \tag{84}$$

Here $\hat{\mathcal{P}}$ is the zero order projector onto the $\omega$-RSM and $\hat{\mathcal{S}}$ is the reduced resolvent of $\hat{\mathcal{L}}(s)$ at $i\omega$ which obeys $\hat{\mathcal{S}}\hat{\mathcal{P}} = \hat{\mathcal{P}}\hat{\mathcal{S}} = 0$ and $\hat{\mathcal{S}}(\hat{\mathcal{L}} - \lambda\mathcal{I}) = (\hat{\mathcal{L}} - \lambda\mathcal{I})\hat{\mathcal{S}} = \mathcal{I} - \hat{\mathcal{P}}$. We also write (as on page 78 of [207])

$$\left[\hat{\mathcal{L}}(s)\right]_{\hat{\mathcal{P}}(s)} = i\omega\hat{\mathcal{P}} + s[\hat{\mathcal{L}}_1]_{\hat{\mathcal{P}}} + \mathcal{O}(s^2\|\hat{\mathcal{L}}_2\|). \tag{85}$$

Since $\hat{\mathcal{L}}_1 = -i[H^{(1)}, \circ]$ we can see that $\hat{\mathcal{L}}_1$ generates unitary dynamics, and thus its projection to the $\omega$-RSM, $[\hat{\mathcal{L}}_1]_{\hat{\mathcal{P}}}$ also generates unitary dynamics and thus has purely imaginary eigenvalues which are semi-simple. By the reduction arguments in Section II.3 [207], we can deduce that the eigenvalue $\lambda(s)$ is twice differentiable,

$$\lambda(s) = i\omega + i\lambda_1 s + \lambda_2 s^2 + o(s^2), \tag{86}$$

corresponding to $p = 1$ in Thm 3. □

## I Proof of Theorem 4

**Theorem.** *If there is a full rank stationary state $\rho_\infty$ (i.e. no zero eigenvalues, invertible) and the commutant $\{H, L_\mu, L_\mu^\dagger\}' = c\mathbb{1}$, then there are no purely imaginary eigenvalues of $\hat{\mathcal{L}}$ and hence no stable synchronization. If this holds in the leading order, there are no almost purely imaginary eigenvalues and no metastable synchronization.*

*Proof.* Suppose, that there exists a state $\rho$ with purely imaginary eigenvalue $i\omega$. Since $\tilde{\rho}_\infty$ has full rank, by Thm. 1 we can write $\rho = A\rho_\infty$ where $[H, A] = \omega A$, $[L, A] = [L^\dagger, A] = 0$ and $\hat{\mathcal{L}}[\rho_\infty] = 0$. We then see that $A^\dagger A$ and $AA^\dagger$ both belong to the commutant $\{H, L_\mu, L_\mu^\dagger\}'$, and so

$$A^\dagger A = c_1\mathbb{1}, \quad AA^\dagger = c_2\mathbb{1}. \tag{87}$$

Taking the trace trivially gives $c_1 = c_2 = c$. Notice that $c \neq 0$, since $c = \frac{1}{d}\text{Tr}(A^\dagger A) = \frac{1}{d}\|A\|^2$, and so $c = 0$ would correspond to $A = 0$. Now we compute,

$$\text{Tr}\left(A^\dagger[H, A]\right) = \text{Tr}\,\omega A^\dagger A \tag{88a}$$

$$\text{Tr}\left(A^\dagger HA - A^\dagger AH\right) = \omega\,\text{Tr}\,A^\dagger A \tag{88b}$$

$$0 = \omega\,\text{Tr}(A^\dagger A). \tag{88c}$$

Hence $\omega = 0$. Consequently, no non-zero purely imaginary eignenvalue can exist. $\qquad\square$

## J   Conditions For Meta-stable Synchronization

In Sec. 2.2 it was stated that in the long time limit the dynamics of an observable $\hat{O}$ are described by

$$\langle \hat{O} \rangle(t) = \sum_k e^{t\lambda_k} \langle\langle \hat{O} | \rho_k \rangle\rangle \langle\langle \sigma_k | \rho_0 \rangle\rangle. \tag{89}$$

This is because the purely imaginary eigenvalues of $\hat{\mathcal{L}}$ are semi-simple and thus have trivial Jordan normal form. Since we consider a system of finite size, there is also some $\mu > 0$ such that all eigenvalues $\lambda$ with non-zero real part have $\text{Re}(\lambda) < -\mu$. This Liouvillian gap determines the time period during which transient dynamics occur, and after which the system is synchronized, i.e. the system will be synchronized for $t >> 1/\mu$ provided the relevant criteria of Cor 2 or 3 apply.

When we introduce perturbations, writing

$$\hat{\mathcal{L}}'(s) = \hat{\mathcal{L}} + s\hat{\mathcal{L}}_1 + s^2\hat{\mathcal{L}}_2 + \mathcal{O}(s^2), \tag{90}$$

the eigenvalues vary continuously with the perturbative parameter $s$, and as a result so too will the gap $\mu$. Provided the pertubration is small enough that the gap does not close, i.e $\mu \nrightarrow 0$, in the long time limit we need only consider perturbations to the eigenvalues of $\hat{\mathcal{L}}$ which lie on the imaginary axis. Since these are semi-simple we may diagonalise $\hat{\mathcal{L}}$ over this subspace. Now using the Baker-Campbell-Hausdorff (BCH) formula we can write

$$e^{t\hat{\mathcal{L}}'(s)} = e^{t\hat{\mathcal{L}}} e^{ts\hat{\mathcal{L}}_1 + o(st)}, \tag{91}$$

where $e^{t\hat{\mathcal{L}}}$ is diagonalisable over the subspace of purely imaginary eigenvalues. Thus for $t < \mathcal{O}(1/s)$ the dynamics are determined by $\hat{\mathcal{L}}$ and we can apply the results of Cor. 2 & 3 to $\hat{\mathcal{L}}$ to determine whether meta-stable synchronization occurs.

If the perturbation is of the form

$$\begin{aligned} H(s) &= H^{(0)} + sH^{(1)} + \mathcal{O}(s^2) \\ L_\mu(s) &= L_\mu^{(0)} + \mathcal{O}(s^2), \end{aligned} \tag{92}$$

then by Cor. 5 the eigenvalues of $\hat{\mathcal{L}} + s\hat{\mathcal{L}}_1$ are semi-simple. In this case we can proceed as above and use the BCH formula to write

$$e^{t\hat{\mathcal{L}}'(s)} = e^{t(\hat{\mathcal{L}} + s\hat{\mathcal{L}}_1)} e^{ts^2\hat{\mathcal{L}}_2 + o(s^2 t)}, \tag{93}$$

where now $e^{t(\hat{\mathcal{L}} + s\hat{\mathcal{L}}_1)}$ is diagonalisable. Consequently for $t < \mathcal{O}(1/s^2)$ the dynamics are determined by $\hat{\mathcal{L}} + s\hat{\mathcal{L}}_1$ and we can apply the results of Cor. 2 & 3 to $\hat{\mathcal{L}} + s\hat{\mathcal{L}}_1$ to determine whether meta-stable synchronization occurs.

## K   Application of our Theory to Non-Markovian Systems

This appendix indicates how our theory can be applied to systems where the environment is not Markovian. Let the full system-environment Hamiltonian be $H_{\text{tot}}$, and suppose we can partition the full system-environment into (i) The system we are interested in, (ii) a part of

the environment that can be modelled as Markovian and (iii) a part of the environment which is non-Markovian. We can write this as

$$H_{\text{tot}} = H_{\tilde{S}} + H_E + H_{\tilde{S},E}, \tag{94}$$

where $\tilde{S}$ contains both the system *and* the non-Markovian environment and $E$ contains exclusively the Markovian portion of the environment. Labeling the reduced density operator for the system *and* the non-Markovian environment as $\tilde{\rho}$ we can trace out the Markoivan environemnt to obtain the Lindblad equation for $\tilde{\rho}$,

$$\frac{d}{dt}\tilde{\rho} = -\mathrm{i}[H_{\tilde{S}}, \tilde{\rho}] + \sum_\mu 2L_\mu \tilde{\rho} L_\mu^\dagger - \{L_\mu^\dagger L_\mu, \tilde{\rho}\}. \tag{95}$$

We can now apply the results of the main text to determine that the long-time dynamics of $\tilde{\rho}$ are given by

$$\tilde{\rho}(t) \to \sum_k e^{\mathrm{i}\omega_k t} c_k \tilde{A}_k \tilde{\rho}_{k,\infty}, \tag{96}$$

where $c_k$ are constants determined by the initial conditions and $\omega_k$, $\tilde{A}_k$, & $\tilde{\rho}_{k,\infty}$ are as in Thm. 1. In this form we can study synchronization by directly applying our results to the operators $\tilde{A}_k$, & $\tilde{\rho}_{k,\infty}$. This is illustrated in the example in Sec 6.1.

We can also trace out the non-Markovian environment to obtain the reduced density matrix of the system as

$$\rho(t) \to \sum_k e^{\mathrm{i}\omega_k t} c_k \operatorname{Tr}_{\tilde{E}}\left(\tilde{A}_k \tilde{\rho}_{k,\infty}\right). \tag{97}$$

Unfortunately, it is not possible to draw general conclusions directly about the dynamics of the system without knowing the details of the non-Markovian bath and applying our results to the combined system. If the fine details of the non-Markovian bath were inaccessible or too difficult to study our theory would likely not be generally applicable but this would require a case by case analysis. Other more systematic approaches to non-Markovian generalisations of the Lindblad equation can be found in [208, 209].

## L    Analysis of Additional Examples of Quantum Synchronization

To further demonstrate our theory, we now provide additional analysis of previously studied examples of quantum synchronization. We will focus on the model of two weakly coupled, driven-dissipative spin-1 systems as previously studied in a synchronization setting by [37]. In the absence of external interactions, two coupled spins, labelled $A$ and $B$, evolve according to the Hamiltonian

$$H = \omega_A S_A^z + \omega_B S_B^z + \frac{i\epsilon}{2}\left(S_A^+ S_B^- - S_B^+ S_A^-\right), \tag{98}$$

where for convenience we define the detuning $\Delta = \omega_A - \omega_B$. We then consider the independent interactions of each spin with some external bath which in the absense of spin-spin interactions drives the spins towards their own non-equilibrium steady states. These system-bath interactions are modeled by the Lindblad operators

$$L_{u,j} = \gamma_j^u S_j^+ S_j^z, \ L_{d,j} = \gamma_j^d S_j^- S_j^z, \ \ j = A, B. \tag{99}$$

Using the theory we have developed, we will analyze three examples which were shown by [37] to exhibit quantum synchronization. The first example considers driving the two spins in opposite directions without any detuning, the second example introduces detuning but also takes the *quantum Zeno* limit of large driving, and the third considers driving two detuned spins in opposite directions.

## L.1 Inverted Limit Cycle: Metastable ultra-low frequency anti-synchronization

We first analyse the so called inverted limit cycle. We take $\Delta = 0$ and invert the driving on the two spins so that

$$\gamma_A^u = \gamma_B^d = \gamma, \ \gamma_A^d = \gamma_B^u = \mu, \tag{100}$$

and consider the limiting case $\mu \to 0$. As in [37] the system is initialised in the state $\rho(0) = \rho_A^{(0)} \otimes \rho_B^{(0)}$ where $\rho_j^{(0)}$ is the NESS of spin $j$ in the absence of spin-spin couplings. Thus we can consider this as a quench of the system where the weak spin-spin interaction are instantaneously turned on. We find that in the absence of spin-spin interactions, i.e $\epsilon = 0$, and with $\mu = 0$ the independent spins have degenerate stationary states

$$\begin{aligned}
\rho_A^{(0)} &= p_A |0\rangle_A \langle 0|_A + (1 - p_A) |1\rangle_A \langle 1|_A, \\
\rho_B^{(0)} &= p_B |0\rangle_B \langle 0|_B + (1 - p_B) |-1\rangle_B \langle -1|_B,
\end{aligned} \tag{101}$$

for $p_j \in [0, 1]$. This degeneracy is lifted as soon as $\mu$ becomes strictly positive and we find $p_A = p_b = 1$ in Eq. (101). Thus, to avoid this degeneracy and simplify discussions, in the following we consider $\mu$ infinitesimally small but still strictly positive. Consqeuently we consider both $\mu$ and $\epsilon$ as perturbative parameters while $\omega$, $\gamma$ remain $\mathcal{O}(1)$.

To understand the evolution of the system it is sufficient to consider those eigenstates of $\hat{\mathcal{L}}$ which are excited by $\rho(0)$, and in particular which of these have eigenvalues with small real part compared to $\omega$ and $\gamma$. We find that only 4 eigenmodes with small real parts are excited, with corresponding eigenvalues

$$\begin{aligned}
\lambda = 0, \ &-2\mu + \mathcal{O}(\mu^2, \epsilon^2, \epsilon\mu), \\
&-\mu \pm 2\mathrm{i}\epsilon + \mathcal{O}(\mu^2, \epsilon^2, \epsilon\mu).
\end{aligned} \tag{102}$$

Of these, the relevant eigenvalues for synchronized oscillations are $\lambda_\pm = -\mu \pm 2\mathrm{i}\epsilon + \mathcal{O}(\mu^2, \epsilon^2, \epsilon\mu)$ since they are the only one with a non-zero imaginary part at leading order. We conclude that is that this is an example of ultra-low frequency synchronization since both the real and imaginary parts of $\lambda_\pm$ are $\mathcal{O}(\epsilon, \mu)$. The consequences of this are shown in Figure 6a where we consider the $S_j^z$ observables. We see that the observable appears almost stationary on short time scales, $t \sim \mathcal{O}(1)$. When, in Figure 6b, we consider significantly longer timescales, however, we see the behaviour which we consider metastable synchronization. We further find that the decay rate is inversely proportional to $\epsilon^2$ at the next lowest order, as indicated in Sec. 4.1. Consequently, there is only one power of $\epsilon$ between the decay rate and the frequency of the signal unless $\epsilon$ is sufficiently small that $\mu > \epsilon^2$ in which case the decay rate is propositional to $\mu$. This example demonstrates why we generally discount ultra-low frequency synchronization since it is unfeasible to observe experimentally.

## L.2 Inverted Limit Cycle: Anti-synchronization in the Quantum Zeno Limit

We again consider the same system as above, but now detuned, $\omega_A \neq \omega_B$, and with strong dissipation, $\gamma_A^u = \gamma_B^d = \gamma \gg \omega_j, \epsilon$. We find that the dissipation operator

$$\begin{aligned}
\hat{\mathcal{D}}[\rho] = {}&2S_A^+ S_A^z \rho S_A^z S_A^- + 2S_B^- S_B^z \rho S_B^z S_B^+ \\
&- \{S_A^z S_A^- S_A^+ S_A^z, \rho\} - \{S_B^z S_B^+ S_B^- S_B^z, \rho\}
\end{aligned} \tag{103}$$

has a stationary subspace with 16-fold degeneracy. Thus, when we lift the degeneracy of this subspace by introducing comparatively weak unitary dynamics, we introduce several eigenvalues with $\mathcal{O}(1)$ imaginary parts and $\mathcal{O}(1/\gamma)$ real parts. This leads to metastable dynamics on timescales relevant to the Hamiltonian. In Fig. 7 we see that for the initial state

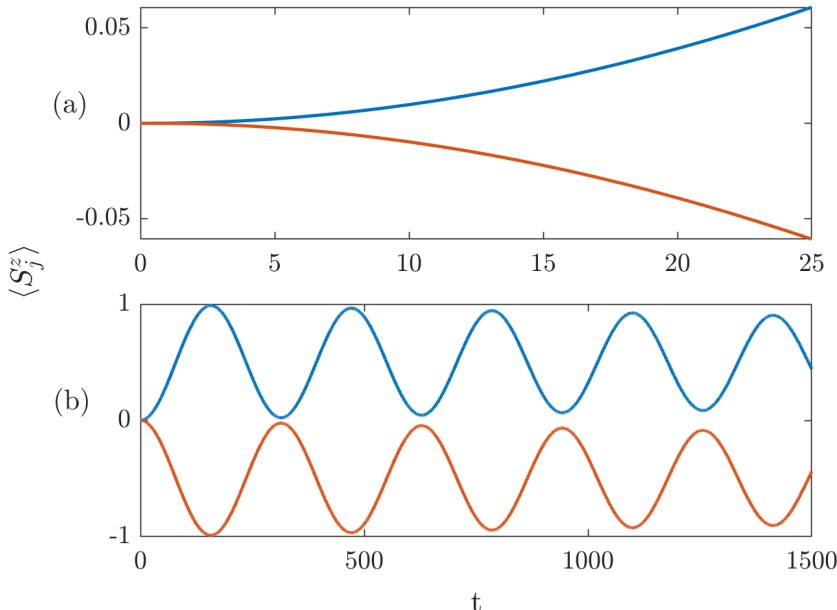

Figure 6: Evolution of $S^z$ observable on sites $A$ (blue) and $B$ (red) for the Spin-1, inverted limit cycle model with $\omega = \gamma = 1$, $\mu = 0.0001$, $\epsilon = 0.01$. In (a) we see over short time periods the observables gradually move away from 0 while in (b) we see that over much longer time scales decaying oscillations can be measured. As per (102) the frequency of these oscillations is $2\epsilon$

$\rho(0) = |0\rangle_A \langle 0|_A \otimes |0\rangle_B \langle 0|_B$ there are clean, synchronized oscillations in the $S^z$ observable. When initialised in this particular state, this metastable anti-synchronization occurs for any $\omega, \epsilon << \gamma$.

Unlike the previous example, we find that this meta-stable anti-synchronisation is robust to initialising the system in arbitrary states. This can be seen by noting that the eigenstates of the dissipation operator, $\hat{\mathcal{D}}$, are coherences between the states

$$|0,0\rangle , \ |1,-1\rangle , \ |0,-1\rangle , \ |1,0\rangle . \tag{104}$$

Under strong dissipation, the system rapidly decays onto the space spanned by these eigenstates before the slower dynamics take over. Since this space spanned by coherences of the above eigenstates is invariant under the transformation $S^z_A \longleftrightarrow -S^B_z$, all dynamics within this space will satisfy $\langle S^z_A(t) \rangle = -\langle S^z_B(t) \rangle$ regardless of the initial state. However, we find that the system is not completely anti-synchronized since if we measure instead the observable $S^x_j$ we find that the measurements on the two sites are now uncorrelated.

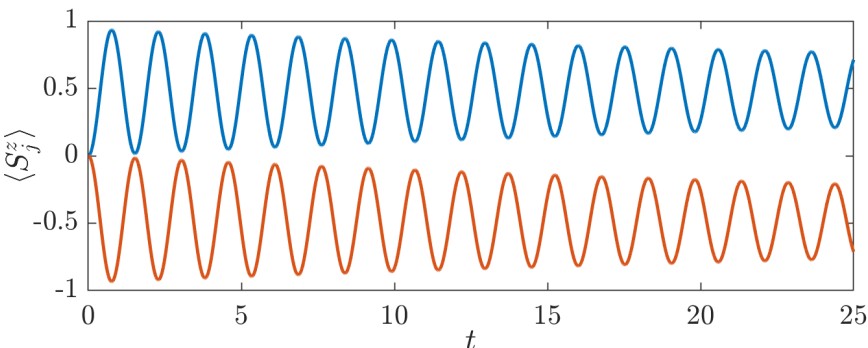

Figure 7: Metastable anti-synchonisation of the inverted limit cycle model in the Quantum Zeno limit for $\gamma = 100$, $\mu = 0$, $\omega_A = 0.5$, $\omega_B = 1.5$ and $\epsilon = 2$ when the system is initialised in $\rho(0) = |0\rangle_A \langle 0|_A \otimes |0\rangle_B \langle 0|_B$. This is a consequence of the unitary dynamics lifting the degeneracy in the stationary subspace of the dissipation. Comparison with Fig. 6 shows that in the Zeno limit the oscillations are now on timescales relevant to the Hamiltonian.

## L.3  Pure Gain or Loss: Stable Limit Cycles, but no Robust, Stable Synchronization

We now set $\gamma_j^d = 0$, $\gamma_A^u \neq \gamma_B^u \neq 0$. In that case we find three proper (density matrix) stationary states,

$$\rho_{1,\infty} = |-1, 0\rangle \langle -1, 0|$$
$$+ \frac{i\epsilon}{\omega_A - \omega_B}(|-1, 0\rangle \langle 0, -1| - |0, -1\rangle \langle -1, 0|), \tag{105}$$

$$\rho_{2,\infty} = |-1, -1\rangle \langle -1, -1|, \tag{106}$$

$$\rho_{3,\infty} = \frac{1}{2}(|0, -1\rangle \langle 0, -1| + |-1, 0\rangle \langle -1, 0|). \tag{107}$$

Solving for the conditions of Th. 1, we may find the $A$,

$$A_1 = a_1 |-1, -1\rangle \langle -1, 0|$$
$$+ i(a_2 - a_3)|-1, -1\rangle \langle 0, -1|, \tag{108}$$

$$A_2 = A_1^T, \tag{109}$$

$$A_3 = -i(a_2 + a_3)|-1, 0\rangle \langle -1, 0|$$
$$+ a_1 |-1, 0\rangle \langle 0, -1|$$
$$- a_1 \frac{a_2 + a_3}{a_3 - a_2} |0, -1\rangle \langle -1, 0|$$
$$+ i(a_2 - a_3)|0, -1\rangle \langle 0, -1|, \tag{110}$$

where $a_1 = 2\epsilon$ and $a_2 = (\omega_A - \omega_B)$, $a_3 = \sqrt{4\epsilon^2 + (\omega_A - \omega_B)^2}$. The corresponding frequencies are, $\omega_1 = \frac{1}{2}(\omega_A + \omega_B - a_3)$, $\omega_2 = \frac{1}{2}(\omega_A + \omega_B + a_3)$, $\omega_3 = a_3$. There is no permutation or generalized symmetry between sites $A$ and $B$ and so although we do have persistent oscillations and a limit cycle the sites are not robustly synchronized as seen in Fig. 8.

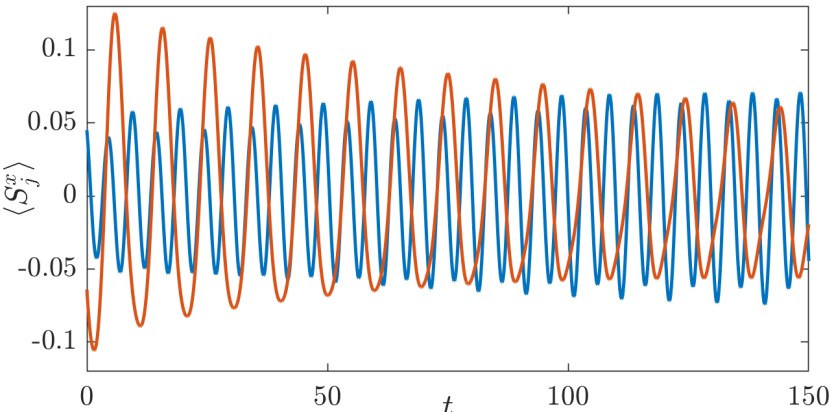

Figure 8: Evolution of $S^x$ observable on sites $A$ (blue) and $B$ (red) for the Spin-1, pure gain model. We have non-zero detuning and interaction, $\Delta, \epsilon \neq 0$ and asymmetric driving, $\gamma_A^u \neq \gamma_B^u$. Initialising the system in a random state, we see that while persistent oscillations occur the two sites do not synchronize. This can be understood through the absence of generalised symmetry between sites $A$ and $B$.

## M  A General Framework for Constructing Models which Exhibit Quantum Synchronization

In Sec. 6.2 we studied models of many-body systems which exhibit quantum synchronization. In particular, the Fermi Hubbard model with spin-agnostic heating hinted at a general framework for constructing more elaborate models which exhibit this behaviour. To understand this, we make the following observations:

(i) In the absence of a magnetic field and potential, $B_j = \epsilon_j = 0$, the Hubbard model has a symmetry group $G = SU(2) \times SU(2)/\mathbb{Z}^2$, coming from the independent spin and $\eta$ symmetries [210].

(ii) Further, the representation of these symmetries are permutation invariant, that is

$$[S^\alpha, P_{j,k}] = [\eta^\alpha, P_{j,k}] = 0, \quad \alpha = x, y, z \,.$$

(iii) Introduction of the magnetic field, $BS^z$, which corresponds to the unique element of the Cartan subalgebra of the spin-$SU(2)$ symmetry, breaks the spin symmetry. Consequently, the remaining elements, $S^+$, $S^-$ of the spin-$\mathfrak{su}(2)$ algebra become dynamical symmetries

(iv) Choosing a unital set of Lindblad operators, $L_\mu$, from the complementary $\eta$ symmetry grantees that $[L_\mu, S^\pm] = 0$ so that the spin operators are strong dynamical symmetries as required for Thm. 2. Also, since the choice of Lindblad operators do not all commute with the $\eta^\alpha$ operators, there can be no further strong dynamical symmetries. Finally, since these strong dynamical symmetries have complete permutation invariance, the synchronization is robust, as per Cor. 3.

These principles can be applied more widely to models with more elaborate symmetry structures in order to guarantee quantum synchronization, such as the generalized $SU(N)$ models in Sec 6.3

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
