# Peer review of "Algebraic Theory of Quantum Synchronization and Limit Cycles under Dissipation"

_SciPost Physics, doi:SciPost Phys. 12, 097 (2022)_

## Round 2 · Referee Report · Anonymous (Referee 1) · 2021-6-1

Strengths

1 - Elegant and illuminating theory of quantum synchronisation using the Liouvillian formalism including perturbative results for metastable synchronisation, linking many results of the field together.
2 - Thorough introduction into the field of synchronisation, strong overview of both the field's history as well as recent results in the literature.
3 - Manuscript provides several examples for theoretical results provided, good experimental relevance.
4 - Extensive appendix provides detailed proofs of all theorems and corollaries including additional examples which provide further context to the main results.

Weaknesses

(see Report for details)
1 - Problematic clarity in several places due to, e.g., use of highly specialised terminology.
2 - Introduction and discussion of Liouvillian formalism is flawed.
3 - Structure of manuscript not streamlined optimally in some places.

Report

--- General Remarks ---

This manuscript provides a novel theory of quantum synchronisation using the Liouvillian formalism. By associating the emergence of synchronisation with imaginary eigenvalues of the Liouvillian and symmetries of the dynamical generator, the authors are able to prove general results which are applicable to a wide variety of physical systems. In particular the importance of the evolution's unitality is interesting and noteworthy. The authors both embed their findings in a detailed discussion of the field's history as well as illustrate their main results with several experimentally relevant examples.

Unfortunately, the text suffers from a very high level of specialised language. Furthermore several technical aspects are often not explained in a clear manner. While the results themselves are intriguing and certainly warrant publication, it would be highly desirable if the manuscript could be made more accessible and if the discussion of some key concepts could be corrected and/or clarified. In the following I will provide further details on a section-by-section basis.

--- Section 1 ---

  • I very much appreciate the detailed overview of the field, starting from classical considerations and moving to the quantum domain. However, reading through pages 4-6 I became a little lost as to the relevance of many of the discussed topics to the subject of the paper - quantum synchronisation. For example the listing of classes of synchronicity are quickly supplanted with the definitions from Sec. 2. It might be helpful to keep the introduction slightly more streamlined to the paper's central theme and remind the reader of the implication for the quantum domain regularly and clearly.

  • On page 8 the sentence "Indeed, even when the stationary state can be found analytically diagonalizing it, as is required to find its support, is still an open problem." needs to be entirely reworked.

--- Section 2 ---

  • There is a brief mention on page 9 regarding "alternative cases" which refers to synchronised signals that differ by a phase or a scale factor. This discussion should be expanded to make clear in how far and in which places the results of this work need to be altered to account for them.

  • The remark on page 10 regarding the fact that all definitions are understood as equalities up to exponentially small terms in time is very important and could be emphasised more (e.g. by giving it its own paragraph). I also feel like calling it "common-place in physics" is overstating how customary it actually is.

  • The Lindblad formalism is not well introduced and explained on page 11. First of all it is not true that all smooth one-parameter families of completely-positive and trace-preserving (CPTP) maps have Lindblad form. This can be seen by considering a general, potentially non-Markovian evolution which originates from partial tracing a unitary evolution of a larger system (which is still a convolution of continuous operations). It is the semigroup property that is crucial which is unfortunately not mentioned at all when refering to the original work by Lindblad. As such the Lindblad formalism is not the most general form of a quantum evolution (even though it is correct that as a point-to-point map for a single point in time, CPTP maps are general). This misconception is present in several parts of the paper which is particularly puzzling since in Sec. 5.1 the authors very explicitly make use of a non-Markovian environment themselves. While I do agree that the Lindblad formalism is a sensible framework for this study, its generality should not be overstated and in particular the role of non-Markovian environments should be discussed since they are explicitly exploited in the manuscript after all. Finally it should be mentioned that the Lindblad formalism can also be employed for strong system-environment coupling and not only in the weak-coupling limit as the authors claim.

  • The use of "trivial Jordan form" on page 12 seems overly convoluted and I only was able to understand the meaning by referencing the appendix. In fact, "(block-)diagonalisable" would be a nomenclature that is more easily accessible to a wide audience I think.

--- Section 3 ---

  • I was not quite able to understand in how far Theorem 1 neccessarily implicates a non-equilibrium steady-state. If I consider a Liouvillian that is trivial, e.g. the zero operator, then clearly all states are eigenstates with eigenvalue zero which would fall under the conditions for the theorem. However no state is a non-equilibrium steady state in this case since expectation values of all observables are constant in time because no actual evolution takes place. I seem to be missing an argument as to the sufficiency of the theorem in this and related special cases.

  • On page 14 the sentence "In order to have \omega \neq 0 we can see by taking the trace of equation (15) that A must not be unitary but additionally must be traceless.". To my understanding of the previous discussion A still can be unitary but does not have to be. As such I suspect in this sentence it should be "A need not be unitary" instead of "A must not be unitary".

  • The discussion at the end of Sec. 3.1 as to the commensurability was not quite clear to me. The authors claim there is a danger in having "too many incommensurable purely imaginary eigenvalues" and state that "it should be understood that considerations regarding commensurability must be made in addition to the results provided.". I do not quite understand the scope to which these statements should be interpreted since the phrasing used here is very vague.

--- Section 4 ---

  • While I do like the general discussion in Secs. 4.1 - 4.3 it took me several re-reads to properly place them into the big picture of the section. It might be better to start with the general statements in 4.3 and Theorem 3 (to my understanding this theorem should capture all other cases since \omega = 0 is not excluded from its scope) and move from there to the more specialised cases of 4.1 and 4.2 and see how they emerge from Theorem 3.

  • On page 17 and 18 I have trouble to understand the use of the nomenclature "explicity breaking the exchange symmetry". If I understand it well this corresponds to a vanishing anticommutator of the Liouvillian with the operator of exchange symmetry. While this does indeed break the exchange symmetry, so would any scenario in which the commutator does not vanish and one might even argue that a vanishing anticommutator is a kind of very specific breaking of the symmetry that preserves some structure.

--- Section 5 ---

  • As mentioned above, the fact that a non-Markovian bath is employed for anti-synchronisation seems very important to me and should not be properly put into the bigger scope of the whole paper.

  • The connection between decoherence-free subspaces and synchronisation mentioned at the bottom of page 20 is quite intriguing and might warrant further discussion and/or future study.

  • On page 21 the vanishing anticommutator of A with P_{2,3} is called "antisymmetry" which seems to me more appropriate than the "explicit symmetry breaking" nomenclature used in Sec. 4.

  • The discussion regarding a general framework of quantum synchronization of the model in Sec. 5.2. was very hard to follow. It also uses several concepts that might not be too familiar to most readers with a background in physics (e.g. the Cartan subalgebra).

--- Section 6 ---

  • This section feels somewhat out of place. It might be a good idea to move Theorem 4 and the corresponding discussion to the end of Section 3 where all general results and theorems are collected and discussed.

--- Final Remarks ---

The science presented in the manuscript is intriguing and original. However, there are some flaws that still need to be amended before publication, most notably the clarity of the presentation. Once the authors address these weaknesses in a revision, I would recommend this paper for publication in SciPost Physics.

Requested changes

(see Report for details)
1 - Improve clarity and accessibility of manuscript.
2 - Amend discussion of Liouvillian formalism by, e.g., mentioning the role of non-Markovianity in quantum evolutions, in particular with respect to its direct application in Sec. 5.1.

  • validity: high
  • significance: high
  • originality: high
  • clarity: ok
  • formatting: excellent
  • grammar: good

Author:  Berislav Buca  on 2021-11-12  [id 1932]

(in reply to Report 1 on 2021-06-01)
Category:
answer to question
reply to objection

We thank the referee for their careful consideration of our work and detailed comments. Here we respond to their requested changes directly and outline how we have improved the manuscript accordingly.

Section 1

1.1. “I very much appreciate the detailed overview of the field, starting from classical considerations and moving to the quantum domain. However, reading through pages 4-6 I became a little lost as to the relevance of many of the discussed topics to the subject of the paper - quantum synchronization. For example the listing of classes of synchronicity are quickly supplanted with the definitions from Sec. 2. It might be helpful to keep the introduction slightly more streamlined to the paper's central theme and remind the reader of the implication for the quantum domain regularly and clearly.”

As suggested, this section has been streamlined. This is along with other modifications to the introduction, as suggested by the other referees.

1.2 “On page 8 the sentence ``Indeed, even when the stationary state can be found analytically diagonalizing it, as is required to find its support, is still an open problem.'' needs to be entirely reworked”

This sentence has now been replaced with ``Although in some cases analytic tools can be used to find non-equilibrium steady states, it remains an open problem to efficiently find the support of a generic stationary state.''

Section 2

2.1 “There is a brief mention on page 9 regarding ``alternative cases'' which refers to synchronized signals that differ by a phase or a scale factor. This discussion should be expanded to make clear in how far and in which places the results of this work need to be altered to account for them.”

We have now indicated what needs to be altered in Cor. 2 of Sec 3.2 to apply our results to phase synchronization. This clarifies how case-by-case considerations need to be made for similar generalized modes of synchronization (i.e. differing by a scale factor/constant).

2.2 “The remark on page 10 regarding the fact that all definitions are understood as equalities up to exponentially small terms in time is very important and could be emphasized more (e.g. by giving it its own paragraph). I also feel like calling it ``common-place in physics'' is overstating how customary it actually is.”

As suggested, we have made this remark its own paragraph to emphasize it more.

2.3 “The Lindblad formalism is not well introduced and explained on page 11. First of all it is not true that all smooth one-parameter families of completely-positive and trace-preserving (CPTP) maps have Lindblad form. This can be seen by considering a general, potentially non-Markovian evolution which originates from partial tracing a unitary evolution of a larger system (which is still a convolution of continuous operations). It is the semigroup property that is crucial, which is unfortunately not mentioned at all when referring to the original work by Lindblad. As such the Lindblad formalism is not the most general form of a quantum evolution (even though it is correct that as a point-to-point map for a single point in time, CPTP maps are general). This misconception is present in several parts of the paper which is particularly puzzling since in Sec. 5.1 the authors very explicitly make use of a non-Markovian environment themselves. While I do agree that the Lindblad formalism is a sensible framework for this study, its generality should not be overstated, and in particular the role of non-Markovian environments should be discussed since they are explicitly exploited in the manuscript after all. Finally it should be mentioned that the Lindblad formalism can also be employed for strong system-environment coupling and not only in the weak-coupling limit as the authors claim.”

We have now amended this discussion in our manuscript to avoid confusion. We had initially inferred the semi-group property by assuming time-homogeneity and smoothness, but we have now made it explicit that the semi-group property is required. We also did not intend to claim that the Lindblad formalism can only be used for weak systems. Indeed this is why we explain that the Lindblad master equation can describe a 1-parameter family of CPTP maps with the semi-group property. We only intended to point out that the intuition regarding jump operators is only valid for weak coupling.

We have also explained that our theory is not limited to Markovian environments as we can, in principle, use the Hamiltonian to describe the full system-environment interactions. This has been explained more clearly in a new appendix.

2.4 “The use of ``trivial Jordan form'' on page 12 seems overly convoluted and I only was able to understand the meaning by referencing the appendix. In fact, ``(block-) diagonalizable'' would be a nomenclature that is more easily accessible to a wide audience I think.”

As suggested, this wording has changed to be more accessible.

Section 3

3.1 “I was not quite able to understand in how far Theorem 1 necessarily implicates a non-equilibrium steady-state. If I consider a Liouvillian that is trivial, e.g. the zero operator, then clearly all states are eigenstates with eigenvalue zero which would fall under the conditions for the theorem. However, no state is a non-equilibrium steady state in this case since expectation values of all observables are constant in time because no actual evolution takes place. I seem to be missing an argument as to the sufficiency of the theorem in this and related special cases.”

We work with the definition that a density operator $\rho_\infty$ is a NESS if $\mathcal{L}[\rho_\infty] = 0$. This is the definition that is generally used in the literature as most non-trivial systems only have a small number of such states, and thus a general initial state will evolve until it reaches some superposition of NESS's. We agree with the referee that this terminology does not make complete sense in some special cases, as the referee has pointed out, but we reiterate that this is the general definition taken by the community and is the most straightforward terminology for our work.

With regards to the question of applying our theorems to trivial Liouvillian case, the conditions of Thm 1. are still necessary and sufficient for there to exist a purely imaginary eigenvalue, although they are satisfied trivially for any operator $A$ with eigenvalue $\lambda = 0$ since $ H = L_\mu = 0$. In this case, there will be no $A$ which satisfies the conditions for $\lambda \ne 0$. However, we emphasize that this system would not be classified as synchronized in our work since our definitions require there to be some form of dynamics.

3.2 “On page 14 the sentence ``In order to have $\omega \neq 0$ we can see by taking the trace of equation (15) that $A$ must not be unitary but additionally must be traceless.'' To my understanding of the previous discussion $A$ still can be unitary but does not have to be. As such I suspect in this sentence it should be ``$A$ need not be unitary'' instead of ``$A$ must not be unitary''

Consider the following
\begin{align*}
&[H, A] = \omega A\\
&A^{-1}H A - H = \omega {I} \ \ \ \text{Assuming $A$ is invertible} \\
&0 = \omega \text{Tr}(I) \ \ \ \text{Taking trace}
\end{align*}
where $I$ is the identity. Thus we cannot have $A$ invertible and $\omega \ne 0$. Hence since $A$ cannot be invertible, it certainly cannot be unitary. This manipulation has now been included in the manuscript to avoid confusion.

3.3 “The discussion at the end of Sec. 3.1 as to the commensurability was not quite clear to me. The authors claim there is a danger in having ``too many incommensurable purely imaginary eigenvalues'' and state that ``it should be understood that considerations regarding commensurability must be made in addition to the results provided.'' I do not quite understand the scope to which these statements should be interpreted since the phrasing used here is very vague.”

The discussion regarding commensurability has been clarified and made into its own subsection at the end of section 3.

Section 4
4.1 “While I do like the general discussion in Secs. 4.1 - 4.3 it took me several re-reads to properly place them into the big picture of the section. It might be better to start with the general statements in 4.3 and Theorem 3 (to my understanding this theorem should capture all other cases since $\omega = 0$ is not excluded from its scope) and move from there to the more specialized cases of 4.1 and 4.2 and see how they emerge from Theorem 3.”

The referee is correct that Thm. 3 does capture the case $\omega = 0$. However, the sections were presented in this order to \emph{avoid} the reader considering the case of 4.1 somehow emerging from Thm. 3. The two cases should be thought of a physically distinct: 4.1 takes a system without long-lived oscillations and shows that ultra-low frequency dynamics can be induced by perturbation, while 4.3 takes a system with long-lived oscillations and shows that under perturbation, the oscillations are stable to first order. We have made this distinction more clear in the text. The ideas were also presented in this manner to clearly accredit the work on the $\omega = 0$ case to Macieszczak et al.

4.2 “On page 17 and 18 I have trouble to understand the use of the nomenclature ``explicitly breaking the exchange symmetry''. If I understand it well, this corresponds to a vanishing anticommutator of the Liouvillian with the operator of exchange symmetry. While this does indeed break the exchange symmetry, so would any scenario in which the commutator does not vanish and one might even argue that a vanishing anticommutator is a kind of very specific breaking of the symmetry that preserves some structure.”

As suggested in a later comment, we have now used the term ``anti-symmetric'' to indicate the specific symmetry breaking we are considering.

Section 5

5.1 “As mentioned above, the fact that a non-Markovian bath is employed for anti-synchronization seems very important to me and should not be properly put into the bigger scope of the whole paper.”

We have now explained more precisely how our theory can be used for both Markovian and non-Markovian environments as per the comment above.

5.2 “The connection between decoherence-free subspaces and synchronization mentioned at the bottom of page 20 is quite intriguing and might warrant further discussion and/or future study.”

This is indeed an intriguing connection which we intend to explore further in future work.

5.3 “On page 21 the vanishing anticommutator of $A$ with $P_{2,3}$ is called ``antisymmetry'' which seems to be more appropriate than the ``explicit symmetry breaking'' nomenclature used in Sec. 4.”

We have updated Sec. 4 to use this more appropriate terminology.

5.4 “The discussion regarding a general framework of quantum synchronization of the model in Sec. 5.2. was very hard to follow. It also uses several concepts that might not be too familiar to most readers with a background in physics (e.g. the Cartan subalgebra).”

We have now moved the explanation of the more general framework to the appendices to lighten the main text and make it more accessible to a broader readership without losing any detail.

Section 6

6.1 “This section feels somewhat out of place. It might be a good idea to move Theorem 4 and the corresponding discussion to the end of Section 3 where all general results and theorems are collected and discussed.”

We have now moved section 6 to before discussing examples where we agree it fits more naturally.

---

## Round 2 · Referee Report · Anonymous (Referee 2) · 2021-6-21

Strengths

1 - novel, well motivated conjecture for a definition and measure of synchronization of multipartite open quantum systems 2-Clear and well defined mathematical model and conjectures in terms of the Lindblad master equation framework 3-instructive set of well worked out examples
4- comprehensive but concise introuction with a very helpful presentation of the state of the art

Weaknesses

1- relation to some older work (Haken) on master slave dynamics or atomic dipole synchronization e.g. in lasers or a bit vague 2- definition seems a bit restrictive in particular for applications on coupled systems of different physical nature, where hamonics of a synchronization frequency could appear in different subsystems 3-connection between phase diffusion and metastable synchronization could be worked out better

Report

The paper well meets the quality standards and scope of the journal

Requested changes

1- the seminal works of Haken on synchronization (Synergetics) 50 years ago should be better represented in the inroduction and definition section

( e.g. https://www.researchgate.net/profile/Peter-Tass/publication/301232987_Synchronization_in_networks_of_limit_cycle_oscillators/links/53da7fc80cf2631430c82735/Synchronization-in-networks-of-limit-cycle-oscillators.pdf )

2-to clarify: does phase locking in a rotating basis as often used for coupled atomic dipoles qualify for synronization as time does not appear explicitly any more ? e.g: https://ui.adsabs.harvard.edu/abs/2014APS..DMP.D1022Z/abstract

3-how about the Dicke superadiant qunatum phase transition ? Is there a atom-field sychronization even in the ground state ?

https://www.sciencedirect.com/science/article/abs/pii/0003491673900390

4- how about the appearance of several cycle frequencies in a system ?

  • validity: top
  • significance: high
  • originality: high
  • clarity: high
  • formatting: excellent
  • grammar: excellent

Author:  Berislav Buca  on 2021-11-12  [id 1933]

(in reply to Report 2 on 2021-06-21)
Category:
answer to question
reply to objection

We thank the referee for their consideration of our work and the useful recommendations and questions. Here we respond to their requested changes directly and outline how we have improved the manuscript accordingly.

  1. “The seminal works of Haken on synchronization (Synergetics) 50 years ago should be better represented in the introduction and definition section.”

We have now included a discussion of Haken's work in our introduction.

  1. “To clarify: does phase locking in a rotating basis as often used for coupled atomic dipoles qualify for synchronization as time does not appear explicitly any more? (e.g. https://ui.adsabs.harvard.edu/abs/2014APS..DMP.D1022Z/abstract)”

Our theory can be used to analyze the behaviour of this system in the rotating frame where time dependence has been removed. It allows for the long-time dynamics to be analyzed in this rotating basis precisely as we do for time-homogeneous systems studied in our paper.

  1. “How about the Dicke superadiant quantum phase transition? Is there a atom-field synchronization even in the ground state? (e.g. https://www.sciencedirect.com/science/article/abs/pii/0003491673900390)”

We are not completely clear what the referee means by synchronization \emph{in the ground state}. The synchronization we are considering is explicitly synchronicity of observables in time. If the system is in the ground state, then the various observables will not evolve in time, and so there will be no synchronization of the kind we study.

  1. “How about the appearance of several cycle frequencies in a system?”

Our theory explicitly allows for multiple cycle frequencies through the existence of multiple $A$ operators, each corresponding to different purely imaginary eigenvalues. This has now been made more evident to the reader in the new subsection (3.3) about multiple frequencies and commensurability.

---

## Round 2 · Referee Report · Anonymous (Referee 3) · 2021-6-23

Strengths

1- Timely topic: quantum synchronization in open systems. 2- Useful Liouvillian description, formal approach building on previous works. 3- Broad list of References

Weaknesses

1- Limits of applicability not stated, lack of counter-examples. 2- Novelty needs to be clarified referring to the literature.

Report

Quantum synchronization (QS) is a timely subject and several original approaches and results have been reported in the last years. In this context, this manuscript complements previous work of some of the authors on dynamical symmetries, as in https://doi.org/10.1088/1367-2630/ab60f5, and on the Liouvillian formalism in QS, as in https://doi.org/10.1103/PhysRevA.95.043807. Looking at theorems and examples in comparison with these works, it is not always clear this work novelty. As mentioned, there is a broad list of references (more than 100 cited in the first page), but pertinent citations when discussing what is new are often missing.
The attempt of a formal description is interesting even if limited to some rather specific scenario for QS. Actually a main difficulty in reading this work is that both the abstract and the overview sections do not clearly define the context of applicability of this approach and after a promise of a “general theory” on QS one finds many limitations.
After reading the whole manuscript it appears that it deals with QS of finite dimensional systems, with interaction with the environment described by a specific class of master equations, not applicable in a master-slave scenario, nor in presence of significantly different subsystems, nor to phase synchronization. Actually the analysis is restricted to the case of identical subsystems (Sect 3.2) and to weak deviations from this scenario (perturbed Liouvillian, Sect 4).

The manuscript revised following the requested changes would be suitable for publication in this journal.

Requested changes

1-The main change this manuscript will benefit from is a clear initial statement about the relevance/applicability of the described framework: the authors should identify (ideally avoiding a technical language) the systems and the kind of synchronization that can be formally described by the framework described in Sections 3 and 4. The abstract claim “ […] no comprehensive theory has been found [on quantum synchronization]. We give such a general theory” should be replaced by a fair statement on the specific reported results. See also 5-. 2- Section 1.1.1 should focus on the relevant synchronization context and properly refer to it. For instance, the authors claim that ref 123 (well-known book on synchronization) considers “synchronization to be a purely periodic phenomenon” when there is even a dedicated chapter on synchronization in chaotic systems. It is also not correct that “In contrast to identical synchronization, phase synchronization does require periodic motion in order to meaningfully define a relative phase difference between the two subsystems.” This is not the case, see for instance https://doi.org/10.1103/PhysRevLett.81.321 On the other hand, the authors long discussion about getting identical synchronization by rescaling different system observables is not really insightful. 3- The scenario of metastability of section 4 is a form of known transient synchronization http://dx.doi.org/https://doi.org/10.1007/978-3-030-31146-9_6 or Ref 43, but no clear connection is discussed. 4- Complete synchronization has been already proposed in https://doi.org/10.1038/s41598-019-56468-x This previous work should be acknowledged when presenting this concept. 5- Referring to 36 and 37, the authors claim that synchronization between spin ½ systems is possible because of non-Markovianity. On the other hand, many works also cited in this manuscript deal with synchronization between spins ½ systems in the Markovian case. Do the authors framework predicts that no stable synchronization can be achieved there? What would “fail” if the authors considered a system of only 2 instead of 3 identical units and a common bath under Markovian dissipation? Mentioning/discussing cases where this formalism cannot be applied while synchronization has been reported would be as useful as the already included examples, adding strength to this work.

Minor comments: -The classification presented in Section 1.1.1 is a bit misleading, mixing the form of synchronization (e.g. in phase or amplitude) with the systems configuration (autonomous vs driven). Also, the very first definition of synchronization in the abstract is not the generally accepted one. Same criticism for the definition in sect.3.2 “Recall, that the crucial feature of quantum synchronization is that the various parts of the subsystem lock into the same phase, frequency and amplitude.” Synchronization is a broader phenomenon. -The physical ground of the strong coupling case described in Eq 21 should be commented and related to possible microscopic derivation. Also the assumption after eq 25 of L_1 anticommuting with the permutation operator should be clarified.

  • validity: high
  • significance: high
  • originality: good
  • clarity: good
  • formatting: excellent
  • grammar: excellent

Author:  Berislav Buca  on 2021-11-12  [id 1934]

(in reply to Report 3 on 2021-06-23)
Category:
answer to question
reply to objection

Please see attached file for our reply.

Attachment:

Reply_to_Referee_3.pdf

---

## Round 4 · Referee Report · Anonymous · 2021-12-2

Strengths
see my first report
Weaknesses
see my first report
Report
see my first report
Requested changes
see my first report
Author: Berislav Buca on 2022-01-12 [id 2095]
(in reply to Report 3 on 2021-12-10)We thank the referee for their careful consideration of our work and detailed comments. Here we respond to their requested changes directly and outline how we have improved the manuscript accordingly.
Point 1:
"The first is about the claimed impossibility to synchronize two ½ spins. The authors argument stands on (their) unusual definition of synchronization for which two temporal signals:
\begin{equation*}
\cos(t) \quad \& \quad a + \cos (t)
\end{equation*}
would not be synchronized, due to the presence of an offset.
Generally synchronization is considered looking at the dynamical part of signals and any constant off-set is actually neglected. Beyond the authors choice of “terminology”, they should notice that actually these signals would be perfectly synchronized also using the same synchronization indicator of their Eqs. 1 and 2."
Firstly, we must emphasise that we have nowhere claimed that it is impossible synchronize two spin-1/2s. In our response to the referee's previous comments, we provided an example of two spin-1/2s under a common Markovian bath and explained that under our stricter definition, these two spins do not anti-synchronize but that \emph{``under a weaker definition of synchronization we could consider this system to be (anti-)synchronized''.
We chose to work with the strictest definition of synchronization in our work to avoid additional technicalities in the main theorems as these technicalities offer little additional insight. In our previous revision we already made indication of where additional considerations could be made to obtain weaker notions of synchronization. In our latest version we have included a new subsection dedicated to this discussion in order to make these ideas clearer. We have also explicitly pointed out to the reader that our definitions are stricter that the Pearson correlation indicator.
Point 2:
"The authors answer:
“The claim that we cannot treat subsystems that are not identical is incorrect. We now give an explicit example in Sec. 6.2.1 ...”
Maybe there was a misunderstanding, as I was referring specifically to synchronization.
Indeed “stable quantum synchronization” is addressed in Section 3.2 providing sufficient conditions in two corollaries. Both of them assume the exchange/permutation symmetry between sites, the first in the Liouvillian and the second in the operator A, rather strong symmetries indeed at the basis of my comment.
The authors do not refer to this in their answer but highlight the example in Sec. 6.2.1, actually already in the previous version of the manuscript. This is a rather elaborated model and in the present form it does not clarify this point.
It is clear that the global $S^+$ and $S^-$ are permutation invariant, while the Hamiltonian is not.
On the other hand, in order to appreciate the claimed generality, could the authors specify the local operators $O_j$ that would be identically synchronized in this inhomogeneous model?
It would be also useful to include the definition of $c_j$ in the “agnostic” local ladders (38) and (39)."
We now recognise our misunderstanding, and apologise to the referee. Our framework can only treat systems where the local Hilbert spaces of each site have the same dimension - i.e. we cannot consider synchronizing a spin-1/2 with a spin-1. We have made this clearer in our revised manuscript, and explicitly pointed out that our definition for robustness would no longer be applicable for sites with different local Hilbert space dimensions. We have also outlined in our conclusions that extending our theory for studying synchronisation between subsystems with different local Hilbert spaces is an active focus of future work.
We now recognise our misunderstanding, and apologise to the referee. Our framework can only treat systems where the local Hilbert spaces of each site have the same dimension - i.e. we cannot consider synchronizing a spin-1/2 with a spin-1. We have made this clearer in our revised manuscript, and explicitly pointed out that our definition for robustness would no longer be applicable for sites with different local Hilbert space dimensions. We have also outlined in our conclusions that extending our theory for studying synchronisation between subsystems with different local Hilbert spaces is an active focus of future work.
Finally, we thank the referee for identifying a typographical mistake, Eqs. (38) and (38) should read
\begin{align*}
L^{-}_{j}&=\gamma^-_{j} c_{j, \uparrow}c_{j, \downarrow}\\
L^+_{j}&=\gamma^+_{j} c^\dagger_{j, \uparrow}c^\dagger_{j, \downarrow}
\end{align*}
Point 3:
"As a note, the newly added theorem 4 would be really less obscure clarifying the meaning of the “commutant” proportional to the Identity."
This has been clarified by a footnote.

---

## Round 4 · Referee Report · Anonymous · 2021-12-5

Report
The presentation of the manuscript has significantly improved in this revision - the structure is now much cleaner, the introduction sharper and the presentation of the main theorems and proofs is more approachable to a non-specialised reader.
In particular, the authors refined their introduction to the Lindblad equation. Nevertheless, I would appreciate if the authors decided for a uniform approach as to whether they focus on the Markovian "spirit" of the master equation or whether they want to emphasise non-Markovian extensions. The authors argue that one may place part of the environment inside the system's Hilbert space to allow for non-Markovian dynamics. However, one might argue that the actual environment represented by the Lindblad operators is still purely Markovian. Note that there also exist alternative extensions to bring the Lindblad formalism to the non-Markovian regime, see e.g. the work by Breuer (Phys. Rev. A 75, 022103) or the work by Head-Marsden and Maziotti (Phys. Rev. A 99, 022109), but in my view these are all extensions to the - in itself Markovian - Lindblad formalism. Although my concern is primarily semantic, I think it is nonetheless worthwhile to stick as close as possible to the established literature when it comes to discussing the Lindblad master equation framework.
In any case, the discussion in Appendix K is much appreciated. Nevertheless, I think it would be important to include the central conclusion "[...] it is not possible to draw general conclusions directly about the dynamics of the system without knowing the details of the non-Markovian bath and applying our results to the combined system." in the main text.
Finally, I would recommend another thorough proofreading of the text to amend the remaining grammatical issues. For example, in the introduction it should be "characterize" in the sentence "These results provide a general theory for studying quantum synchronization and characterizes the phenomenon." Another example is at the end of the introductory text for 4.1 where I suppose that by "We also remark that generically there is only one order of s different between the decay rate and the oscillation frequency." the authors probably mean "[...] the order of s only differs by one between the decay rate [...]", i.e., it is not about the existence of a different order but rather that the order itself is different, if I understand it well.
In conclusion, barring the two minor remarks mentioned above, I can now recommend this manuscript for publication in SciPost Physics.
Author: Berislav Buca on 2022-01-12 [id 2094]
(in reply to Report 1 on 2021-12-05)
We thank the referee for their careful consideration of our work and detailed comments. Here we respond to their requested changes directly and outline how we have improved the manuscript accordingly.
Point 1:
"The authors refined their introduction to the Lindblad equation. Nevertheless, I would appreciate if the authors decided for a uniform approach as to whether they focus on the Markovian "spirit" of the master equation or whether they want to emphasise non-Markovian extensions. The authors argue that one may place part of the environment inside the system's Hilbert space to allow for non-Markovian dynamics. However, one might argue that the actual environment represented by the Lindblad operators is still purely Markovian. Note that there also exist alternative extensions to bring the Lindblad formalism to the non-Markovian regime, see e.g. the work by Breuer (Phys. Rev. A 75, 022103) or the work by Head-Marsden and Maziotti (Phys. Rev. A 99, 022109), but in my view these are all extensions to the - in itself Markovian - Lindblad formalism. Although my concern is primarily semantic, I think it is nonetheless worthwhile to stick as close as possible to the established literature when it comes to discussing the Lindblad master equation framework.
In any case, the discussion in Appendix K is much appreciated. Nevertheless, I think it would be important to include the central conclusion "[...] it is not possible to draw general conclusions directly about the dynamics of the system without knowing the details of the non-Markovian bath and applying our results to the combined system." in the main text."
As advised by the referee, we have now restricted the discussion in the main text so that it focuses exclusively on the Markovian regime. We have then amended our appendix slightly and included the references suggested by the referee.
Point 2:
"Finally, I would recommend another thorough proofreading of the text to amend the remaining grammatical issues. For example, in the introduction it should be "characterize" in the sentence "These results provide a general theory for studying quantum synchronization and characterizes the phenomenon." Another example is at the end of the introductory text for 4.1 where I suppose that by "We also remark that generically there is only one order of s different between the decay rate and the oscillation frequency." the authors probably mean "[...] the order of s only differs by one between the decay rate [...]", i.e., it is not about the existence of a different order but rather that the order itself is different, if I understand it well."
We thank the referee for their careful reading of our manuscript. As advised, we have thoroughly proof read the manuscript again and eliminated the remaining grammatical issues found.

---

## Round 4 · Referee Report · Anonymous · 2021-12-10

Report
I acknowledge the authors for their revisions and answers. Focusing on the manuscript main content, and beyond presentation refinements, I appreciate the formal presentation of this manuscript and additions to the previous one, but I have to insist on two points.
The first is about the claimed impossibility to synchronize two ½ spins. The authors argument stands on (their) unusual definition of synchronization for which two temporal signals:
cos(t)
a+cos(t)
would not be synchronized, due to the presence of an offset.
Generally synchronization is considered looking at the dynamical part of signals and any constant off-set is actually neglected. Beyond the authors choice of “terminology”, they should notice that actually these signals would be perfectly synchronized also using the same synchronization indicator of their Eqs. 1 and 2.
The second is about the applicability of this framework to non-identical subsystems.
The authors answer:
“The claim that we cannot treat subsystems that are not identical is incorrect. We now give an explicit example in Sec. 6.2.1 ...”
Maybe there was a misunderstanding, as I was referring specifically to synchronization.
Indeed “stable quantum synchronization” is addressed in Section 3.2 providing sufficient conditions in two corollaries. Both of them assume the exchange/permutation symmetry between sites, the first in the Liouvillian and the second in the operator A, rather strong symmetries indeed at the basis of my comment.
The authors do not refer to this in their answer but highlight the example in Sec. 6.2.1, actually already in the previous version of the manuscript. This is a rather elaborated model and in the present form it does not clarify this point.
It is clear that the global S^+ and S^- are permutation invariant, while the Hamiltonian is not.
On the other hand, in order to appreciate the claimed generality, could the authors specify the local operators Oj that would be identically synchronized in this inhomogeneous model?
It would be also useful to include the definition of c_j in the “agnostic” local ladders (38) and (39).
As a note, the newly added theorem 4 would be really less obscure clarifying the meaning of the “commutant” proportional to the Identity.

---

## Round 4 · Author Response

We would like to thank you and the referees for carefully assessing our work. As requested, we have revised our manuscript according to the referees' suggestions, and we have attached a list of changes together with responses to each of the referees' individual comments. We hope they find our improvements acceptable.
However, we ask you to reconsider your recommendation to transfer our work to SciPost Physics Core. All three referees' reports stated that the work was suitable for publication in SciPost Physics subject to revisions. The referees also stated that the work was of 'high' significance. The only report raising criticism with the results and not the presentation is No. 3. They refer to the scope and novelty of our work, e.g. claiming that we can only treat identical subsystems and weak perturbations from that. This is incorrect, and we now give an explicit example with subsystems that are non-perturbatively different. The rest of the criticism stems from differences in terminology, which we have now fully addressed. This further demonstrates the importance of linking different areas of quantum synchronization together, which we provide as noted by Ref. 2, and aligns with one of the acceptance criteria of SciPost Physics. Our results are generally applicable to time-independent quantum master equations as stated before and are entirely novel because they give both the necessary and sufficient conditions for quantum limit cycles and (spontaneous) synchronization. As outlined in our cover letter, our fundamental theoretical results provide a paradigm shift linking various fields of quantum synchronization and non-linear dynamics together. This opens a new research direction, with several follow up works already proposed as evidenced by the preprint accumulating 10 citations since being posted on the arXiv in March: https://ui.adsabs.harvard.edu/abs/2021arXiv210301808B/abstract. Thus we believe that we have firmly met the criteria for publication in SciPost Physics.
Below we reply to each referee individually.
We look forward to your and the referees' reply.
Yours sincerely,
Berislav Buca, Cameron Booker and Dieter Jaksch

---

## Round 4 · List of Changes

Note that we have not listed minor corrections of spelling, grammar and punctuation.
Abstract:
Clarify that we are studying spontaneous synchronisation in time-homogeneous systems.
Section 1
1. The sentence ``Indeed, even when the stationary state can be found analytically diagonalizing it, as is required to find its support, is still an open problem.'' on page 8 has been replaced with ``Although in some cases analytic tools can be used to find non-equilibrium steady states, it remains an open problem to efficiently find the support of a generic stationary state.''
2. Rewording of introduction to clarify that we are considering un-driven spontaneous synchronization.
3. `Streamlining' of Sec. 1.1.1 to only focus on topics relevant to our study. References to Haken's work on Synergetisc have also been included.
Section 2
1. The remark on page 9 regarding exponentially small terms has been made into its own paragraph.
2. On page 11 the phrase ``trivial Jordan form'' has been replaced with ``diagonalizable''.
3. Added a footnote to explain that what we call meta-stable synchronization has also been called transient synchronization in the past.
4. Modification to the introduction of the Lindblad formalism to make clearer how our theory can be applied to both markovian and non-markovian systems.
5. Clarified that our discussion refers to synchronization between identical subsystems.
Section 3
1. The manipulation to show the non-unitarity of $A$ in Thm. 2 has been added.
2. The discussion regarding commensurability has been made into its own subsection and reworded to be clearer.
3. Added an example to indicate the modifications required in order to apply our to phase synchronization.
Section 4
1. Subsection 4.3 we emphasise that $\omega \ne 0$. We also now emphasize the distinction between the cases in 4.1 and 4.3.
2. Change the sentence after Thm. 3 to ``Crucially, this means that at first order in $s$ the perturbed eigenvalues remain purely imaginary.''
3. In Cor. 4 we insert the term ``anti-symmetry'' to make it clearer what sort of symmetry breaking we are considering.
Section 5
1. The discussion regarding constructing more general models exhibiting quantum synchronisation has been moved to the appendices.
Section 6
1. Section 6 has been moved to before discussion of various examples. We have also included an example of when this result can be used following the work of Prosen in Ref. [169].
Appendices
1. The discussion regarding constructing more general models exhibiting quantum synchronisation has been moved to the appendices.
2. A new appendix has been added to explain how our theory can be applied to non-Markovian environments.

---

## Round 5 · Referee Report · Anonymous (Referee 1) · 2022-1-14

Report

I appreciate the authors' further revision to their manuscript. All my concerns and remarks were fully addressed such that I recommend this work for publication in SciPost Physics.

---

## Round 5 · Referee Report · Anonymous (Referee 3) · 2022-1-21

Report

In the revised manuscript most points have been clarified, most notably with the addition of sect 3.4. I am recommending this manuscript for SciPost Physics.

---

## Round 5 · Author Response

Dear Editor,

We would like to thank you and the referees again for carefully assessing our responses and the revised manuscript, and for recommending publication in SciPost Physics subject to minor amendments. We have now made further revisions to our work according to the referees' suggestions, and we have attached a list of changes together with responses to each of the referees' individual comments.

We hope they find our minor amendments acceptable.

Yours sincerely,
Berislav Buca, Cameron Booker and Dieter Jaksch

---

## Round 5 · List of Changes

1. In Sec. 2.1 we have explicitly pointed out to the reader that our definitions are stricter that the Pearson correlation indicator.
  2. We have added a new subsection 3.4 to discuss in more detail adaptations of our results to weaker notions of synchronization.
  3. A footnote has been added to Theorem 4 defining the commutant of a set.
  4. References have been added to appendix K as suggested by referee 1.

---

## Editorial Decision

published